



# An assessment of tropopause characteristics of the ERA5 and ERA-Interim meteorological reanalyses

Lars Hoffmann[1] and Reinhold Spang[2]

[1]Jülich Supercomputing Centre, Forschungszentrum Jülich, Jülich, Germany
[2]Institut für Energie- und Klimaforschung (IEK-7), Forschungszentrum Jülich, Jülich, Germany

**Correspondence:** Lars Hoffmann (l.hoffmann@fz-juelich.de)

**Abstract.** The tropopause layer plays a key role in manifold processes in atmospheric chemistry and physics. Here we compare the representation and characteristics of the lapse rate tropopause according to the definition of the World Meteorological Organization (WMO) as estimated from European Centre for Medium-Range Weather Forecasts (ECMWF) reanalysis data. Our study is based on ten-year records (2009 to 2018) of ECMWF's state-of-the-art reanalysis ERA5 and its predecessor

ERA-Interim. The intercomparison reveals notable differences between ERA5 and ERA-Interim tropopause data, in particular on small spatiotemporal scales. The monthly mean differences of ERA5 minus ERA-Interim tropopause heights vary between $-300\,\mathrm{m}$ at the transition from the tropics to the extratropics (near $30°\mathrm{S}$ and $30°\mathrm{N}$) to $150\,\mathrm{m}$ around the equator. Mean tropopause temperatures are mostly lower in ERA5 than in ERA-Interim, with a maximum difference of up to $-1.5\,\mathrm{K}$ in the tropics. Monthly standard deviations of tropopause heights of ERA5 are up to $350\,\mathrm{m}$ or $60\,\%$ larger than for ERA-Interim.

Monthly standard deviations of tropopause temperatures of ERA5 exceed those of ERA-Interim by up to $1.5\,\mathrm{K}$ or $30\,\%$. The occurrence frequencies of double tropopause events in ERA5 exceed those of ERA-Interim by up to $25$ percentage points at mid latitudes. We attribute the differences between the ERA5 and ERA-Interim tropopause data and the larger, more realistic variability of ERA5 to improved spatiotemporal resolution and better representation of geophysical processes in the forecast model as well as improvements in the data assimilation scheme and the utilization of additional observations in ERA5. The

improved spatiotemporal resolution of ERA5 allows for a better representation of mesoscale features, in particular of gravity waves, which affect the temperature profiles in the upper troposphere and lower stratosphere and thus the tropopause height estimates. We evaluated the quality of the ERA5 and ERA-Interim reanalysis tropopause data by comparisons with COSMIC and MetOp Global Positioning System (GPS) satellite observations as well as high-resolution radiosonde profiles. The comparison indicates an uncertainty of the first tropopause for ERA5 (ERA-Interim) of about $\pm150\,\mathrm{m}$ to $\pm200\,\mathrm{m}$ ($\pm250\,\mathrm{m}$) based

on radiosonde data and $\pm120\,\mathrm{m}$ to $\pm150\,\mathrm{m}$ ($\pm170\,\mathrm{m}$ to $\pm200\,\mathrm{m}$) based on the coarser resolution GPS data at different latitudes. Consequently, ERA5 will provide more accurate information than ERA-Interim for future tropopause-related studies.

## 1 Introduction

The transition region between the troposphere and stratosphere plays a key role in atmospheric chemistry and dynamics. Accurate knowledge on the location and temperature of the tropopause comes into play in research topics like stratosphere-





troposphere exchange and mixing events between tropospheric and stratospheric air masses in the extra-tropical transition layer (e. g., Gettelman et al., 2011; Boothe and Homeyer, 2017), transport of water vapor from the troposphere into the stratosphere (Holton et al., 1995; Fueglistaler et al., 2009) and related effects on ozone (Anderson et al., 2012; Robrecht et al., 2021), or the formation of cirrus and convective ice clouds in the lowermost stratosphere (Spang et al., 2015; Zou et al., 2020, 2021). Furthermore, considering the impact of global warming on the temperature structure and general circulation of the troposphere

and stratosphere, long-term changes of the tropopause are also a considered as an indicator of climate change (Seidel et al., 2001; Santer et al., 2003a,b; Sausen and Santer, 2003; Seidel and Randel, 2006).

Various definitions of the tropopause are discussed in the scientific literature (Hoinka, 1997). The definition by the World Meteorological Organization (WMO, 1957) is most commonly for the computation of the lapse rate tropopause and gives robust results for a variety of data sets such as temperature profiles obtained from radiosondes, remote sensing measurements as well

as general circulation models. The lapse rate tropopause yields the sharpest gradient of stability and chemical transitions from the troposphere into the lower stratosphere (Pan et al., 2004; Gettelman et al., 2011). Furthermore, Pan et al. (2018) showed for the tropics, that the lapse rate tropopause describes the transition layer more precisely than the cold point tropopause, taken the chemical tropopause from tracer-tracer correlations of in situ measurements as a reference (e. g., Zahn and Brenninkmeijer, 2003).

Uncertainties in the determination of the lapse rate tropopause may limit the insights scientists can achieve with their analyses. For example, the work of Pan and Munchak (2011), Spang et al. (2015), and Zou et al. (2020) came to different results when trying to quantify the amount of cirrus clouds in the lowermost stratosphere using tropopause information derived with different methods from reanalysis data sets. In their studies, the uncertainty of the tropopause height was a limiting factor, which made a final decision about the amount and potential influence of ice clouds forming in the lower stratosphere very

demanding. Furthermore, in situ measurements frequently show that the boundary between a descending stratospheric intrusion and tropospheric air, as identified by potential vorticity and trace gas constituents, is often several kilometers below the lapse rate tropopause (Homeyer et al., 2010). Especially for analyses based on model data, the vertical resolution may not be sufficient to resolve the stability structure. Improved vertical resolution of reanalysis data would be extremely useful, as these data sets are frequently used for the characterization of the meteorological conditions during airborne campaigns or for driving

chemical transport models for global analyses.

For the evaluation of reanalysis-based tropopause information, vertical high resolution radiosonde and Global Positioning System (GPS) radio occultation temperature profiles are the first choice. Especially, data products like the US High Vertical Resolution Radiosonde Data (HVRRD) (e. g., Wang and Geller, 2003) include a large amount of the temperature fluctuations caused by gravity waves and other mesoscale processes by using improved vertical resolution compared to standard radiosonde

products available from most meteorological stations. Another valuable tropopause data set with excellent vertical resolution and nearly global coverage is available from GPS satellite receivers, delivering temperature soundings with fine accuracy (e. g., Kursinski et al., 1997). Although both kinds of data are part of the assimilation process in established reanalysis data products (HVRRD only with reduced vertical resolution), HVRRD and GPS profiles still provide a good opportunity to quantify uncertainties in the lapse rate tropopause determination from reanalysis data.



Meteorological reanalyses that optimally combine information from both, observations and forecast models, provide comprehensive records of weather and climate changes over time (Kalnay et al., 1996; Dee et al., 2011; Gelaro et al., 2017; Hersbach et al., 2020). Fujiwara et al. (2017) provide an overview on many of the state-of-the-art meteorological reanalyses and on the Stratosphere-troposphere Processes And their Role in Climate (SPARC) Reanalysis Intercomparison Project (S-RIP), which compared reanalysis data sets using a variety of key diagnostics. Although reanalysis data sets may not provide

the same vertical resolution and accuracy as reference data sets such as radiosonde and GPS observations, reanalysis data are a valuable source to study the characteristics and long-term changes of the tropopause on a global scale (Manney et al., 2014; Xian and Homeyer, 2019; Tegtmeier et al., 2020).

The main aim of this study is an intercomparison of the characteristics and representation of the WMO tropopause as derived from ERA5 (Hersbach et al., 2020) and ERA-Interim (Dee et al., 2011) reanalysis data of the European Centre for

Medium-Range Weather Forecasts (ECMWF). In between ERA-Interim and ERA5, there is nearly a decade of developments of the Integrated Forecasting System (IFS) model and the observational data processing and assimilation schemes that are used to produce the reanalyses. In particular, ERA5 provides improved spatiotemporal resolution, allowing for better simulations of mesoscale processes such as gravity waves, convective updrafts, and other processes in the troposphere and stratosphere. Our analysis shows that these improvements of ERA5 over ERA-Interim, in particular the gravity wave-induced temperature

fluctuations, result in significant changes of the tropopause characteristics derived from the data.

In Section 2, we introduce the data and methods used in this study, including brief descriptions of the key features of the ERA5 and ERA-Interim reanalyses (Sect. 2.1), the method used to estimate tropopause height, pressure, and temperature (Sect. 2.2), and the HVRRD and GPS reference data used for evaluation (Sect. 2.3). Section 3 provides the results, i. e., selected examples of ERA5 and ERA-Interim tropopause data (Sect. 3.1), a comparison of 10-year zonal means and standard deviations

of tropopause heights (Sect. 3.2), and occurrence frequencies of double tropopauses (Sect. 3.3). Sections 3.4 and 3.5 focus on the influence of gravity waves and convective updrafts on the tropopause. In Sect. 3.6, we discuss lapse rate statistics and the robustness of tropopause height estimates. Sections 3.7 and 3.8 show comparisons of the tropopause data from the reanalyses with the reference data. Finally, Sect. 4 presents the discussion and conclusions of this study.

## 2    Data and methods

### 85    2.1    The ERA5 and ERA-Interim meteorological reanalyses

In this study, we present a comparison of tropopause characteristics as derived from ECMWF's ERA5 and ERA-Interim reanalyses. The ERA-Interim reanalysis (Dee et al., 2011) was produced using ECMWF's Integrated Forecast System (IFS) Cycle 31r2, which was released in 2006. The reanalysis was produced using 4-d variational analysis with a 12-hour analysis window. The ERA-Interim data are provided for 0, 6, 12, and 18 UTC for the time period from January 1979 to August 2018.

The horizontal resolution of the data is $\sim$79 km ($T_L$255 spectral grid). ERA-Interim covers 60 model levels from the surface up to 0.1 hPa (about 65 km of altitude). We retrieved the ERA-Interim data on a $0.75° \times 0.75°$ longitude-latitude grid and on all model levels from ECMWF's meteorological archive system.





ECMWF's fifth-generation reanalysis, ERA5 (Hersbach et al., 2020), is produced using the IFS Cycle 41r2 as released in 2016. ERA5 improves upon ERA-Interim in various aspects. A major improvement of ERA5 compared to ERA-Interim is its

much higher spatial and temporal resolution. Atmospheric data are available with a horizontal resolution of ∼31 km ($T_L$639 spectral grid). We retrieved the data on a $0.3° × 0.3°$ longitude-latitude grid from ECMWF's meteorological archive, which is equivalent to the spectral resolution of the forecast model. The ERA5 data are provided on 137 hybrid sigma-pressure levels in the vertical, with the top level located at 0.01 hPa (about 80 km of altitude). ERA5 provides hourly estimates of various atmospheric, terrestrial, and oceanic climate variables. In addition to improved resolution, the representation of various

tropospheric and stratospheric processes was enhanced in ERA5 (Hennermann and Berrisford, 2018; Hoffmann et al., 2019). ERA5 is currently under production and will finally cover the time period from January 1950 to present.

In order to process the ECMWF meteorological data with our existing codes, we first interpolated the ERA5 and ERA-Interim temperature data from model levels to pressure levels using the Climate Data Operators (CDO, Schulzweida, 2014) before the tropopause heights were estimated. For this vertical interpolation, the number of the target pressure levels and their

spacing was chosen to correspond to the original ECMWF model levels, using ECMWF's $a$ and $b$ coefficients for the L137 (ERA5) and L60 (ERA-Interim) model level definitions (ECMWF, 2021), respectively, and by assuming a constant surface pressure of 1013.25 hPa. Transferring the ECMWF data from model levels to pressure levels introduces small interpolation errors. However, based on tests comparing ERA5 and ERA-Interim tropopause data derived from pressure level data with independent estimates based on model level data (Appendix A), we found that the interpolation errors are small and can mostly

be neglected for the present study.

The vertical resolution of the ECMWF reanalysis products in the upper troposphere and lower stratosphere region is of particular interest regarding the accuracy of the estimation of the tropopause height and other tropopause parameters. For the data sets considered here, the vertical layer depths at 5 to 20 km of altitude vary between 0.3 to 0.4 km for ERA5 and 0.5 to 1.4 km for ERA-Interim (e. g., Hoffmann et al., 2019, Fig. 1). However, in our analysis, a cubic spline interpolation of the

reanalysis temperature profiles was applied to refine the vertical grid, i. e., the tropopause heights are reported on a finer vertical grid than the reanalysis data. This approach is considered particularly important for ERA-Interim, having much lower vertical resolution than ERA5, in order to allow for a fair comparison of ERA5 and ERA-Interim data. More details on the cubic spline approach are provided in Sect. 2.2 and Appendix B.

## 2.2 Determination of tropopause heights from reanalysis data

To avoid any misinterpretation, as discussed by Maddox and Mullendore (2018), we repeat the exact wording of the definition of the thermal lapse rate tropopause provided by the World Meteorological Organization (WMO, 1957): "(a) The first tropopause is defined as the lowest level at which the lapse rate decreases to 2°C/km or less, provided also the average lapse rate between this level and all higher levels within 2 km does not exceed 2°C/km." We also assess the WMO second tropopause, also referred to as double tropopause, which is defined as "(b) If above the first tropopause the average lapse rate between any level and

all higher levels within 1 km exceeds 3°C/km, then a second tropopause is defined by the same criterion as under (a). This





tropopause may be either within or above the 1 km layer." Therefore, the tropopause is a thermodynamic gradient stratification layer, marking the separation between the troposphere below and the stratosphere above.

In order to estimate the tropopause heights from the reanalysis data, we first interpolated temperatures and geopotential heights from their given pressure levels to a finer vertical grid. Our refined grid had 100 m grid spacing in log-pressure altitudes,
which is about 3 to 4 times denser than the vertical resolution of ERA5 in the upper troposphere and lower stratosphere. Cubic spline interpolation was applied for vertical interpolation, as it is considered to provide a more realistic representation of the shape and fluctuations of real temperature profiles (Zhou et al., 2001; Bell and Geller, 2008; Liu et al., 2010; Peevey et al., 2014; Spang et al., 2015). Cubic spline interpolation is particularly important for coarser resolution data sets such as ERA-Interim and for the tropics, where steep temperature gradients are present around the tropopause.

We restricted the vertical range for the search of the tropopause to pressure levels between 47 and 530 hPa or 4.5 to 21.5 km in log-pressure altitude. If the algorithm failed to identify a tropopause in that vertical range, a missing value is reported. A proper choice of the lower boundary of the search range is important. On the one hand, the lower boundary needs to be low enough, to avoid missing a particularly low real tropopause. On the other hand, the lower boundary needs to be high enough, so that temperature inversions in the lower to mid troposphere are not falsely identified as the tropopause. We note that other
studies typically selected a level of about 500 hPa or 5 km as the lower boundary for the search of the tropopause (Reichler et al., 2003; Peevey et al., 2014; Xian and Homeyer, 2019), which is consistent with the present study.

The first and second thermal tropopause were determined by following the definition of the World Meteorological Organization (WMO, 1957), as given above. Considering that our input data are given on vertically refined pressure levels, the lapse rate $\Gamma$ is calculated using the hydrostatic equation and the ideal gas law,

$$\Gamma = -\frac{dT}{dz} = -\frac{dT}{dp}\frac{dp}{dz} = \frac{gp}{RT}\frac{dT}{dp}, \tag{1}$$

with temperature $T$, geopotential height $z$, pressure $p$, standard gravity $g$, and specific gas constant of dry air $R$. Equation (1) is solved by a finite differencing scheme,

$$\Gamma_{i,j} = \frac{g}{R}\frac{p_i + p_j}{T_i + T_j}\frac{T_i - T_j}{p_i - p_j}, \tag{2}$$

to calculate the average lapse rate $\Gamma_{i,j}$ between the pressure levels $p_i$ and $p_j$ from the temperatures $T_i$ and $T_j$, respectively.
The average lapse rates between a tropopause candidate level and all upper levels within the specified layer depths are assessed following the WMO criteria.

In order to retain the full resolution of the reanalysis data sets, we determined the tropopause heights for both, ERA5 and ERA-Interim, on the same longitude-latitude grid and for the same time steps at which the original reanalysis data sets are provided. We saved the geopotential height, pressure, and temperature values of the WMO first and second tropopause
for further analysis in daily data files. However, various applications require interpolation of the gridded tropopause data to irregular positions in space and time, e. g., when determining tropopause data for observations. In this study, we applied 3-d linear interpolation in time, longitude, and latitude to determine the tropopause data at any given location and time. If tropopause information is missing in the $2 \times 2 \times 2$ time×longitude×latitude data cube surrounding the given location, nearest





neighbor interpolation is applied as a fall-back option to retain the spatiotemporal coverage of the double tropopause events
and their occurrence frequencies.

## 2.3   Radiosonde and GPS reference data

Two observational data sets are used for the evaluation of the reanalysis tropopause data: (a) radiosonde data and (b) GPS
data, nowadays more generally referred to as Global Navigation Satellite System radio occultation (GNSS-RO) data. As both
data sets were assimilated into the reanalyses, the radiosonde and GPS data do not provide independent validation. However,
reanalyses are influenced by various other observations, e. g., satellite-based radiance measurements, as well as the influences
of the forecast models and the data assimilation procedures. Consequently, a comparison to the radiosonde and GPS reference
data is still a meaningful task to assess accuracy and quality of the reanalysis tropopause data.

Radiosondes provide temperature profiles with high vertical resolution, i. e., much better than the resolution of the spline fit.
Here, we selected the US HVRRD data set (e. g., Wang and Geller, 2003), originally compiled for the analysis of gravity waves
but, like also shown, very valuable for studying the thermal structure of the tropopause (Birner, 2006). The HVRRD stations
are located in the US, including some oversea stations spanning a longitude range from $171°$W to $70°$W and a latitude range
from $14°$S to $71°$N, with typical launch times at noon and midnight. The vertical profiles have a time resolution of 6 s, which
results in a vertical sampling of 25 to 30 m around the tropopause, depending on the updraft velocity of the balloon.

In the following analyses, we used the year 2010 with high profile number density ($\sim$11,000 soundings) for the evaluation.
Although the radiosonde profiles are input for the assimilation system of ECMWF, there is additional independent information
content in the HVRRD data, because the reanalyses are processed with a much lower resolution than available from the
HVRRD observations. Small-scale structures in the temperature profile (c. f., Fig. 1a) cannot be resolved by the assimilation
system, but they would actually influence the tropopause determination. The HVRRD data underwent an internal two-stage
quality control process. Only soundings with the highest data quality between 5 and 22 km in altitude are used to identify the
tropopause, applying similar algorithms as described in Sect. 2.2. However, because of the very high vertical resolution, only
a linear fit was applied, with 20 m vertical sampling. With this approach, we kept the variability in the temperature profile and,
consequently, its effect on the tropopause determination. Further, this procedure bypasses potential numerical problems with
the cubic spline fit for very high resolution radiosonde profiles.

In this study, also GPS measurements are used to evaluate the reanalysis tropopause data. A substantial improvement in the
spatial coverage of GPS soundings became available with the Constellation Observing System for Meteorology, Ionosphere
and Climate (COSMIC) Formosa Satellite Mission 3 (Anthes et al., 2008). The COSMIC system consists of six low-orbiting
satellites, providing spatial coverage of about 2000 soundings per day, evenly spread over the globe. This is about an order of
magnitude higher spatial coverage than former radio occultation missions. In the following, we used GPS profiles for the years
2010 (740,000 profiles) and 2017 (570,000 profiles) for comparison of the reanalysis tropopause data. These two years of data
190   include additional GPS profiles from the European MetOp satellites, which are available at the COSMIC data archive as well.
The data considered here are quality assured. The vertical sampling of the COSMIC data set is 100 m, whereas the vertical
resolution is limited to $\sim$200 m (Schmidt et al., 2005). Due to its high measurement density and nearly equally distributed

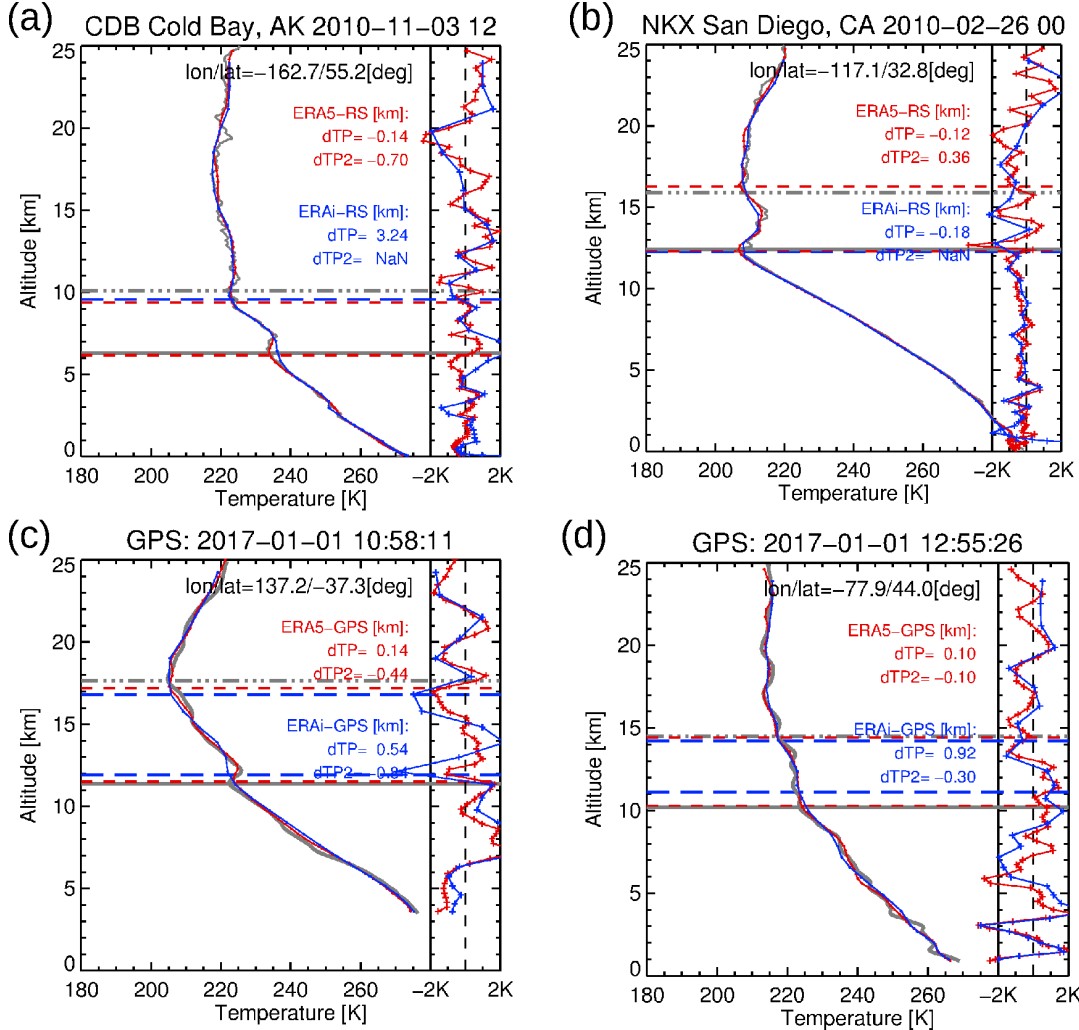

**Figure 1.** Examples of radiosonde (a,b) and GPS (c,d) temperature profiles (gray) in comparison to ERA5 (red) and ERA-Interim (blue) at different location and time. The right hand side profiles show the temperature differences of the reanalyses and the reference data at the vertical resolution of the reanalyses data. Horizontal lines indicate the first and second tropopause height, TPH1 and TPH2, for the radiosondes, GPS, ERA5, and ERA-Interim, respectively.

measurements over the globe, the GPS data set is especially well suited for analyzing the characteristics of the tropopause on the global scale (Son et al., 2011).

195   Four examples of HVRRD and GPS temperature profiles with corresponding nearest ERA-Interim and ERA5 profiles are shown in Fig. 1. The high vertical resolution of HVRRD becomes obvious by the fine wave structures in the temperature profiles in comparison to the superimposed ERA5 and ERA-Interim profiles, which are presented on their native vertical grid. The wave structures are also obvious in the temperature difference profiles shown in Fig. 1. The wave structures are mainly attributed





to gravity waves, which are only partially resolved in the reanalyses data. HVRRD can capture shorter vertical wavelengths than GPS, due to better vertical resolution of the measurements. Both reanalyses cannot resolve the high variability of the radiosonde profiles. However, the smoother shape of temperature profiles of the GPS soundings is better represented by ERA5 than ERA-Interim, which results in a better agreement of the first and second tropopause with GPS data.

## 3 Results

### 3.1 Examples of ERA5 and ERA-Interim tropopause height estimates

As an example, Fig. 2 shows a direct comparison of maps of ERA5 and ERA-Interim tropopause heights on 1 January 2017, 00:00 UTC. For this comparison, we interpolated the ERA-Interim tropopause data to the fine $0.3° \times 0.3°$ longitude-latitude grid of the ERA5 data, in order to calculate the differences of ERA5 minus ERA-Interim on the ERA5 grid (Fig. 2c). A visual inspection of the maps shows that the large-scale structures of the ERA5 and ERA-Interim tropopause are very similar. However, the ERA5 minus ERA-Interim map also reveals many small-scale differences of the tropopause heights, within a typical vertical range of $\pm 1$ km. The tropopause as determined from ERA5 has much more fine structures than the tropopause from ERA-Interim.

Figures 3 and 4 show selected meridional and zonal cross-sections through the ERA5 and ERA-Interim data of Fig. 2. In addition to the WMO first and second tropopause from ERA5 and ERA-Interim, the cross-sections show ERA5 temperature, zonal wind, ice and liquid water content, total column cloud water, and topography. Similar to the maps in Fig. 2, the cross-sections show good agreement of the large-scale structure of the ERA5 and ERA-Interim tropopause, but also reveal more fine structures for ERA5 than for ERA-Interim, in particular in the tropics and at mid latitudes.

A more detailed inspection shows that the larger variability in ERA5 tropopause heights is found in regions where gravity waves from orographic, convective, or jet and storm sources are present. For example, Fig. 4c shows strong mountain wave activity over the Central Andes (near 15°S, 60°W) in the lower stratosphere, which causes substantial variations in tropopause height. As another example, Fig. 4c also shows large tropopause variability in ERA5 associated with tropical convection during the summer monsoon over Northern Australia (15°S, 120 to 150°E). In other regions, where gravity waves or convection are absent, such as the Southern Atlantic or the Indian Ocean, tropopause variability is relatively low. These examples are suggesting that the large variability in ERA5 tropopause heights is associated with larger variability in temperature fields due to mesoscale features such as gravity waves from orographic or convective sources.

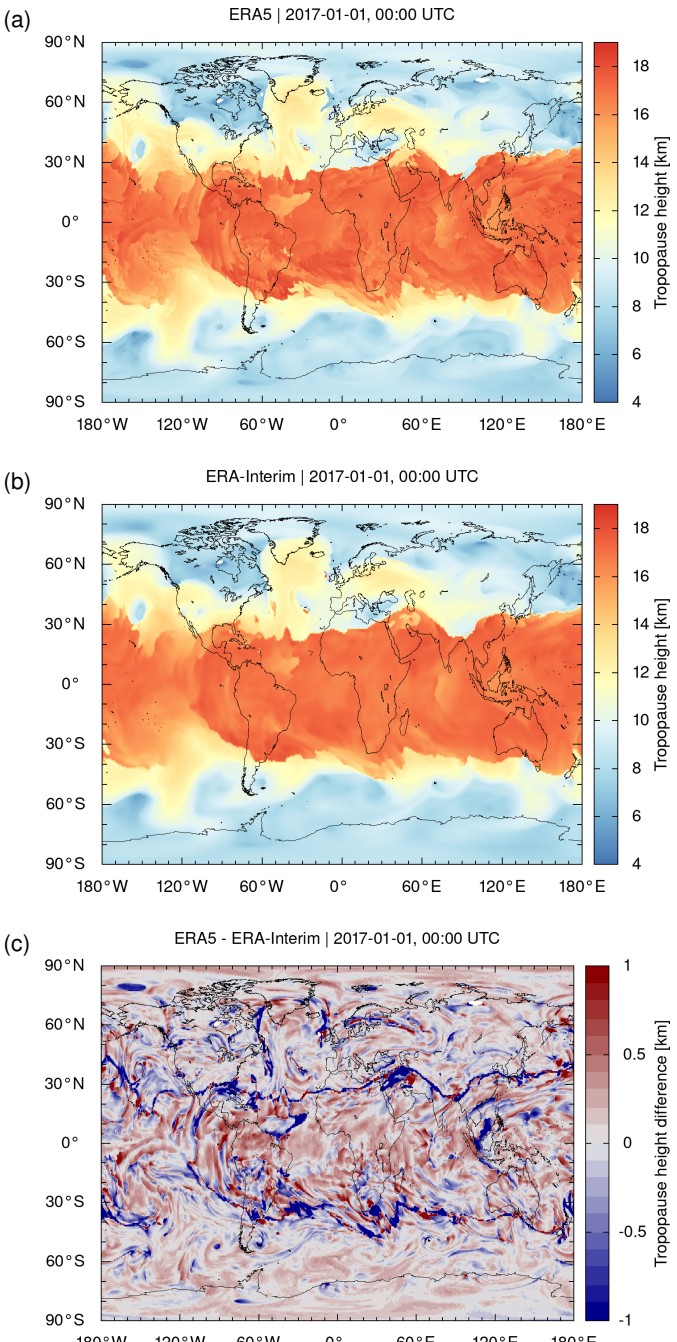

**Figure 2.** Global maps of (a) ERA5 and (b) ERA-Interim WMO first tropopause geopotential heights on 1 January 2017, 00:00 UTC. Differences of ERA5 minus ERA-Interim are shown in (c). The data were sampled on a $0.3° \times 0.3°$ horizontal grid.



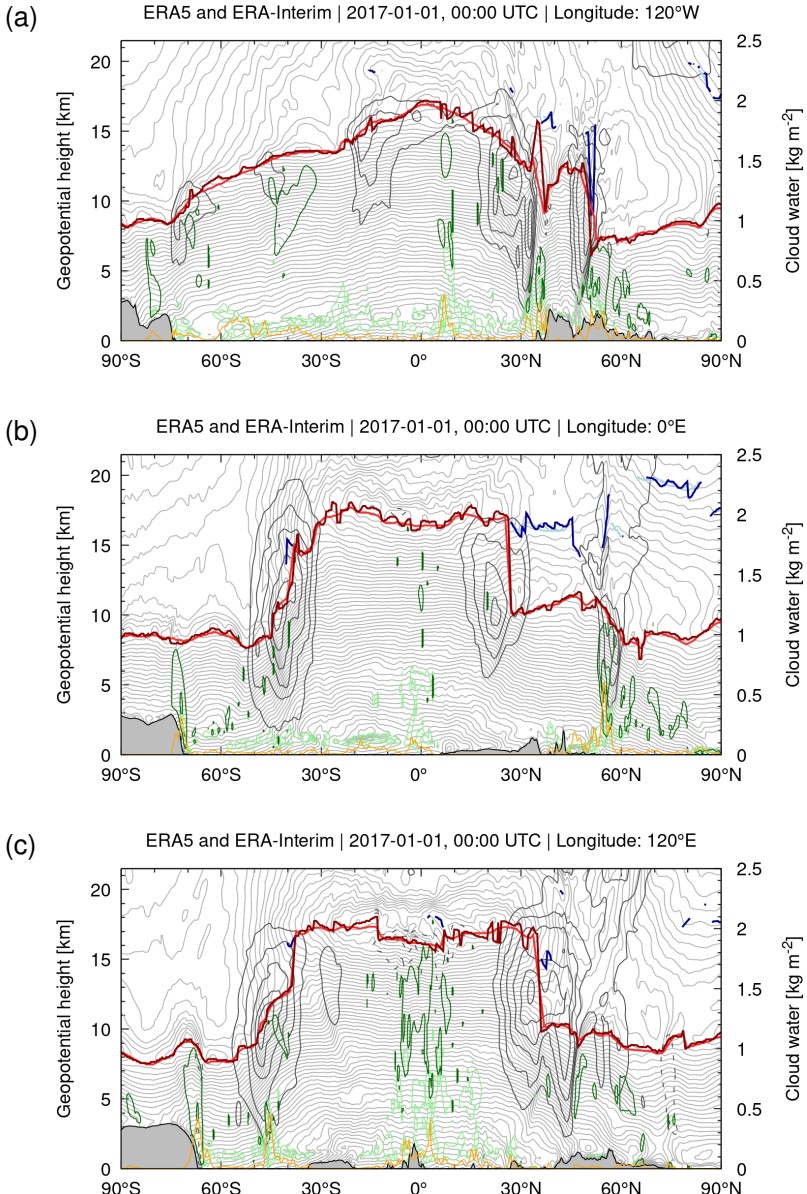

**Figure 3.** Meridional cross-sections of ERA5 (dark red) and ERA-Interim (light red) WMO first tropopause and ERA5 (dark blue) and ERA-Interim (light blue) WMO second tropopause geopotential heights on 1 January 2017, 00:00 UTC at (a) 120°W, (b) 0°E, and (c) 120°E. Light gray curves are ERA5 temperature contours at levels of 2 K. Dark gray curves are ERA5 zonal wind contours at $20, 30, \ldots \mathrm{m\,s^{-1}}$ (solid) and $-20, -30, \ldots \mathrm{m\,s^{-1}}$ (dashed). Dark green and light green curves show contours of the ERA5 cloud ice and liquid water content at levels of 0.01, 0.1, 1, and $10\,\mathrm{g\,kg^{-1}}$, respectively. The orange curve indicates the ERA5 total column cloud water. Gray shaded areas at the bottom of the plots indicate topography. The cross-sections were sampled at $0.3°$ in latitude and $150\,\mathrm{m}$ in height.



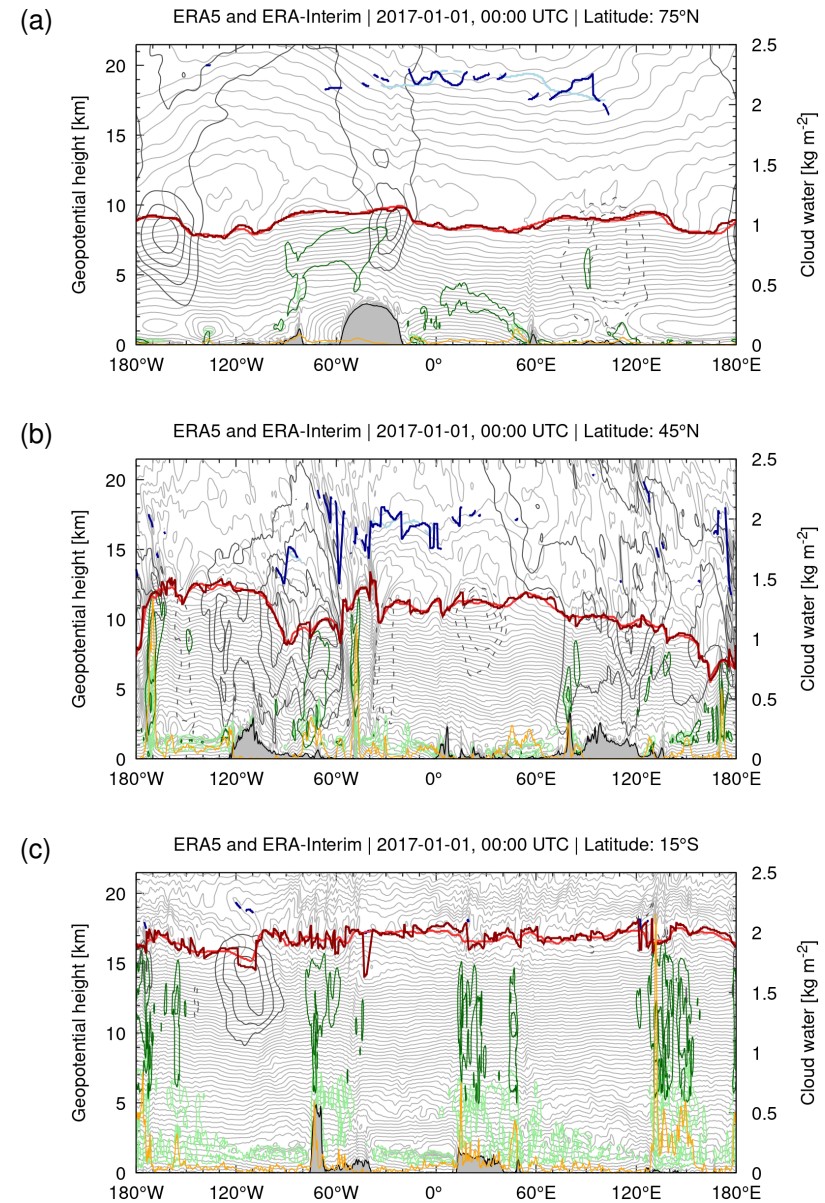

**Figure 4.** Same as Fig. 3, but for zonal cross-sections at (a) 75°N, (b) 45°N, and (c) 15°S. The cross-sections were sampled at 0.3° in longitude and 150 m in height.





## 3.2  Statistical comparison of tropopause heights

In this section, we present a 10-year record of ERA5 tropopause data in terms of monthly mean zonal means and standard deviations. We also discuss the corresponding differences between ERA5 and ERA-Interim. We selected the time period from January 2009 to December 2018 for this analysis, which is close to the end of the production of ERA-Interim data in August 2019. Considering the large amount of ERA5 data, which covers more than $6.3 \times 10^{10}$ vertical profiles for the individual reanalysis grid boxes and time steps, we applied random sampling for data-thinning in the statistical analysis. For each month from January 2009 to December 2018, we created a set of $10^6$ global random samples of locations, onto which we interpolated the ERA5 and ERA-Interim tropopause data. From these samples, we calculated the monthly mean zonal means and standard deviations of the ERA5 and ERA-Interim tropopause geopotential heights and temperatures as well as their differences.

Figure 5 shows the 10-year monthly mean ERA5 data. The mean tropopause height (Fig. 5a) covers a range of 8 and 12.5 km at Southern Hemisphere high latitudes and 8 to 9.5 km at Northern Hemisphere high latitudes. In the tropics, the mean tropopause height varies between 16 and 17 km. At southern and northern mid latitudes, between 30° and 50° of latitude, steep gradients in tropopause heights are found. The corresponding pressure ranges of the tropopause are about 150 to 320 hPa at southern high latitudes, 280 to 330 hPa at northern high latitude, and about 90 to 110 hPa in the tropics (not shown). The zonal mean tropopause temperatures (Fig. 5c) are lowest in the tropics and Southern Hemisphere polar winter (both down to about 190 K) and highest in mid and high latitude polar summer in both hemispheres (up to 225 K).

The monthly zonal standard deviations of the ERA5 tropopause heights (Fig. 5b) are lowest (about 500 m) at polar summer Southern Hemisphere high latitudes and in the tropics. The largest variability (up to 3 km) is found at Southern Hemisphere high latitudes in polar winter and at the transition between the tropics and the extratropics (near 30°S and 30°N). Correspondingly, tropopause temperature variability is as low as 2 to 3 K in the tropics and becomes as large as 9 to 11 K at the transitions from the tropics to the extratropics. Similar to the monthly means, the standard deviations reveal much larger variability of the tropopause height at high latitudes in the Southern Hemisphere compared to the Northern Hemisphere. Qualitatively, the monthly means and standard deviations of the tropopause heights and temperatures found here agree well with earlier studies (Hoinka, 1998; Seidel et al., 2001; Schmidt et al., 2004; Gettelman et al., 2009; Rieckh et al., 2014; Xian and Homeyer, 2019; Tegtmeier et al., 2020), despite the fact that these studies are based on different sets of observations and different ranges of years.

Figure 6 shows the zonal differences of the ERA5 and ERA-Interim tropopause data. The mean differences of the tropopause heights of ERA5 minus ERA-Interim vary between −300 m at the transition from the tropics to the extratropics and 150 m at the Equator (Fig. 6a). The ERA5 tropopause is mostly colder than ERA-Interim, with a maximum difference of −1.5 K at the Equator (Fig. 6b). Similarly, a temperature difference of −0.5 K of the tropical lapse rate tropopause during the years 2002 to 2010 was reported by Tegtmeier et al. (2020). Such temperature differences at the tropical tropopause are physically significant, as they affect the frost point temperatures and control the amount of water vapor entering the stratosphere (Randel et al., 2004; Fueglistaler et al., 2009).





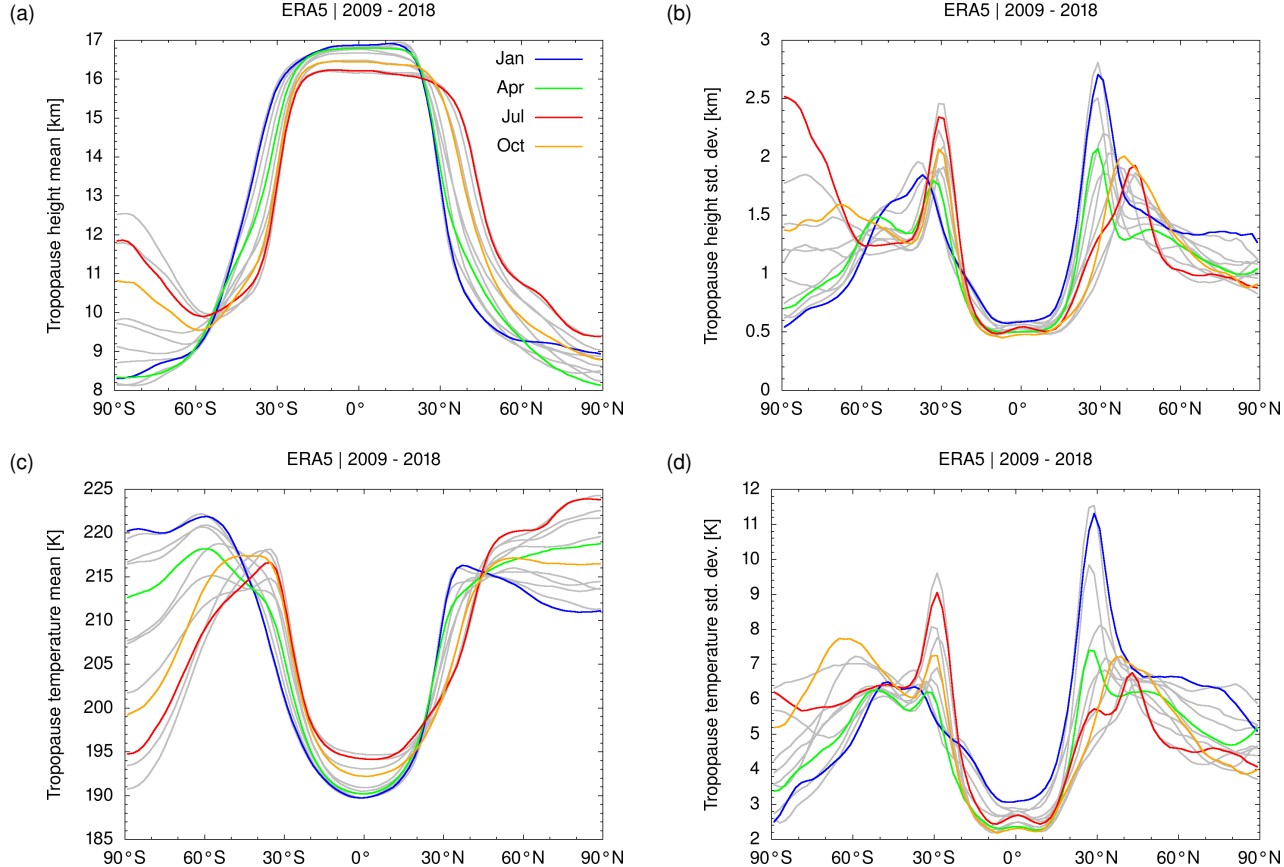

**Figure 5.** Monthly mean zonal means (a,c) and standard deviations (b,d) of WMO first tropopause geopotential height (a,b) and temperature (c,d) based on ERA5. The months of January, April, July, and October are highlighted (see plot key). Gray curves show data for other months. The statistics presented here were calculated for 2° latitude bins from 120 million equally distributed random samples of the ERA5 reanalysis data covering the time period from January 2009 to December 2018.

The standard deviations of the ERA5 tropopause heights are generally larger than those of ERA-Interim. ERA5 standard deviations are about 50 to 100 m larger in the extratropics and about 100 to 150 m larger in the tropics (Fig. 6c). At the

transition from the tropics to the extratropics, the difference of the standard deviations may become as large as 350 m. The absolute differences of the standard deviations found here may appear relatively small. However, the corresponding relative difference of the ERA5 and ERA-Interim standard deviations are in the range of 10 % in the extratropics to 55 % in the tropics (Fig. 6e). This shows that there is notably more variability in the tropopause heights in the ERA5 reanalysis compared to ERA-Interim, in particular in the tropics.

The standard deviations of the tropopause temperatures of ERA5 are typically about 0.5 K larger than those of ERA-Interim (Fig. 6d). Noteworthy exceptions are the transitions from the tropics to the extratropics, with differences as large as 1.4 K, and the Southern Hemisphere high latitudes, with differences of the standard deviations as low as 0.2 K. The relative differences





**Figure 6.** Monthly differences between ERA5 and ERA-Interim geopotential heights (a,c,e) and temperatures (b,d,f) during the years 2009 to 2018. The mean differences between the data sets (a,b) as well as the absolute (c,d) and relative (e,f) differences of the standard deviations are shown. The same bin size and set of samples as for Fig. 5 were used to calculate these statistics.





of the standard deviations of the tropopause temperatures are mostly in the range of 0 to 10 % in the Southern Hemisphere extratropics, 5 to 15 % in the Northern Hemisphere extratropics, and up to 20 to 30 % in the tropics (Fig. 6f). The difference

in variability of the tropopause temperatures between ERA5 and ERA-Interim is not as large as the differences in variability of the tropopause heights, but it is still noteworthy. ERA5 shows larger variability in tropical tropopause temperatures than ERA-Interim.

### 3.3   Comparison of double tropopause occurrence frequencies

The focus of this study is mostly on analyses of the WMO first tropopause. However, we would like to also report a finding

related to the WMO second tropopause as estimated from the reanalysis data. Figure 7 shows maps of the global distributions of the ERA5 and ERA-Interim double tropopauses on 1 January 2017, 00:00 UTC (c. f., Fig. 2). Double tropopauses are mainly found at Northern and Southern Hemisphere mid latitudes in the vicinity of the subtropical jets, marking the transitions from the tropics to the extratropics, which is consistent with the literature (Randel et al., 2007; Añel et al., 2008; Peevey et al., 2014). A direct comparison of the ERA5 and ERA-Interim data in Fig. 7 shows that far more double tropopause are found in

the ERA5 data compared to ERA-Interim. The height distribution of the double tropopause is similar, although ERA5 shows more fine structures than ERA-Interim, like the first tropopause.

Note that double tropopauses are often also detected at winter hemisphere high latitudes. These detections, according to the WMO criteria, need to be considered carefully, though, as they are mostly caused by the strong impact of the polar vortex on the temperature vertical profiles in that region. For example, the meridional and zonal cross-sections in Figs. 3b and 4a indicate

a double tropopause in both, ERA5 and ERA-Interim, at about 70 to 80°N at an altitude of about 18 to 20 km. This height level is about 10 km above the first tropopause and located near the lower boundary of the polar vortex. We therefore consider it as an artifact rather than a real double tropopause event. Although it might be easy to filter such a clear case from the data, the situation becomes more complicated during the course of the winter, when the polar vortex subsidies and its bottom moves closer to the polar tropopause.

Figure 8 shows the monthly mean zonal mean occurrence frequencies of the ERA5 double tropopause events as well as the differences between the occurrence frequencies of ERA5 and ERA-Interim. For ERA5, we found double tropopause occurrence frequencies in the winter months of the corresponding hemispheres as large as 55 % at 30 to 40°S and 70 % at 30 to 40°N (Fig. 8a). Relatively large double tropopause occurrence frequencies (as large as 55 % in the Southern Hemisphere) are also found at high latitudes in the winter season, but these findings need to be considered carefully, due to the impact of the polar vortex

and the near isothermal temperature profiles on the detection of the double tropopause events, as discussed before

Figure 8b shows the differences between the ERA5 and ERA-Interim double tropopause occurrence frequencies. This comparison indicates that ERA5 yields more double tropopause events than ERA-Interim, which we attribute to the improved vertical resolution of the ERA5 data in the lower stratosphere. The differences become as large as 20 percentage points near the mid latitude maxima in both hemispheres. The ERA5 reanalysis shows much more realistic double tropopause occurrence

frequencies in comparison to radiosonde and GPS observations than ERA-Interim. A detailed comparison of double tropopause occurrence frequencies from the reanalyses with radiosonde and GPS reference data is presented in Sects. 3.7 and 3.8.



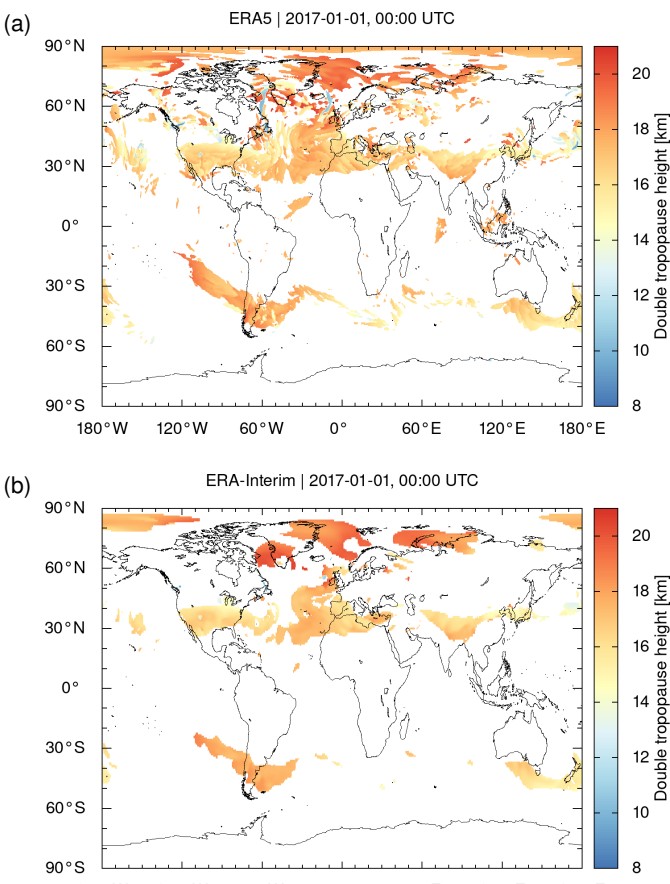

**Figure 7.** Global maps of (a) ERA5 and (b) ERA-Interim double tropopause geopotential heights on 1 January 2017, 00:00 UTC. The data are sampled on a $0.3° \times 0.3°$ horizontal grid. White color indicates areas where no double tropopause was detected. Note that high latitude polar winter detections are artifacts related to isothermal temperature conditions and the influence of the polar vortex.

## 3.4 Tropopause fluctuations due to gravity waves

In this section, we discuss the impact of temperature fluctuations due to gravity waves on the variability of tropopause geopotential heights and temperatures. In particular, as ERA5 provides about 2 to 3 times better horizontal and vertical resolution than ERA-Interim, the IFS model is expected to explicitly resolve gravity waves on finer spatial scales in the ERA5 configuration, which in turn relates to increased variability of the tropopause parameters, as discussed in Sects. 3.1 and 3.2.

In order to determine the temperature fluctuations due to gravity waves, the ERA5 and ERA-Interim temperature fields need to be detrended. A detrending procedure removes background signals due to large-scale temperature gradients and planetary waves. In this study, we applied a horizontal Gaussian high-pass filter for detrending. The temperature background at each pressure level and at each time step of the reanalysis data was estimated independently by smoothing the fully resolved temperature fields with a horizontal Gaussian weighting function with a full width at half maximum (FWHM) of 300 km. The temperature





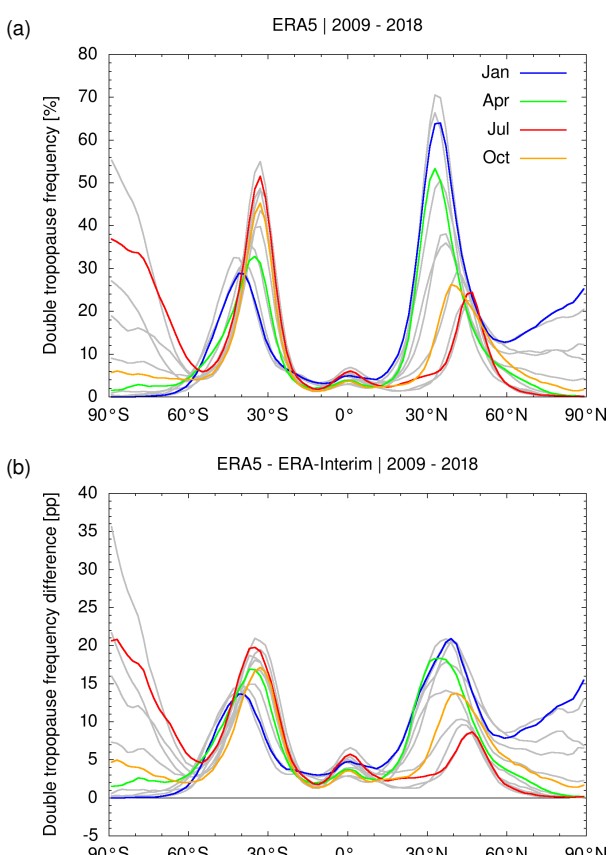

**Figure 8.** Monthly mean zonal mean double tropopause occurrence frequencies in the ERA5 reanalysis (a) and corresponding differences between ERA5 and ERA-Interim (b) in percentage points (pp). The statistics shown here were calculated for $2°$ latitude bins from 120 million global random samples of the ERA5 reanalysis data covering the time period from January 2009 to December 2018.

perturbations due to gravity waves are then estimated by subtracting the smooth background fields from the temperature data. The Gaussian filter has a nominal cut-off frequency of $f_c = 1/(2\pi\sigma)$ with standard deviation $\sigma \approx \text{FWHM}/2.355$, i. e., it will be sensitive to gravity waves with horizontal wavelengths of up to 800 km. An overall minimum for the resolvable horizon-

tal wavelengths is given by the Nyquist frequency, providing a horizontal wavelength of about 60 km for ERA5 and 160 km for ERA-Interim. However, note that the IFS model applies internal filters to maintain numerical stability, i. e., the minimum wavelength that can actually be resolved is larger than the Nyquist frequency.

As an example, Fig. 9 shows ERA5 and ERA-Interim temperature fluctuations due to gravity waves on 1 January 2017, 00:00 UTC derived through the detrending method. As the gravity waves appear on relatively small horizontal scales, we

selected a limited region over South America and the neighboring oceans rather than the global scale for this illustration. The data are shown for the pressure level of 80 hPa (about 18 km of altitude), which is located slightly above the level of the tropical tropopause. Figure 9a shows gravity waves in the ERA5 data propagating upward into the lower stratosphere from


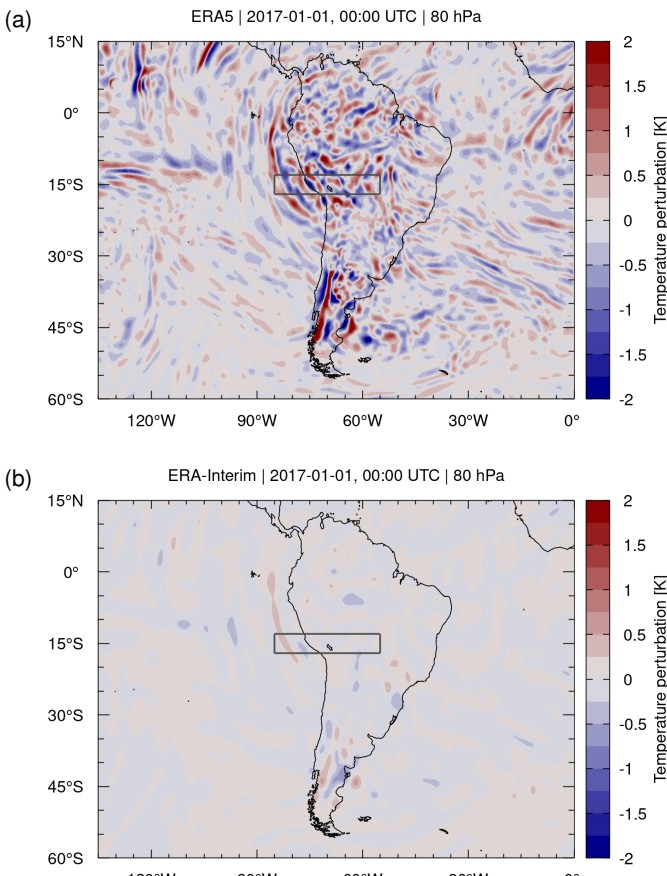

**Figure 9.** Temperature perturbations (contour surface) derived by means of a horizontal Gaussian high-pass filter applied to (a) ERA5 and (b) ERA-Interim reanalysis data on 1 January 2017, 00:00 UTC. Meridional cross-section through the data for the region indicated by the gray box are shown in Fig. 10.

various orographic and non-orographic tropospheric sources. The ERA5 temperature fluctuations related to the gravity waves are typically in a range of $\pm 2$ K at the 80 hPa pressure level. In contrast, the temperature fluctuations in ERA-Interim are either

completely absent or have amplitudes well below $\pm 0.5$ K (Fig. 9b).

Figure 10 shows meridional cross-sections of the ERA5 and ERA-Interim temperature perturbations on 1 January 2017, 00:00 UTC at 15°S and 55 to 85°W. This cross-section traverses the Andes Mountains, an important source of mountain waves, and intersects a hotspot of convective wave activity over Brazil in Southern Hemisphere summer months. The ERA5 cross-section shows coherent gravity wave patterns in the upper troposphere and lower stratosphere. From Fig. 10a, we can

estimate gravity wave vertical wavelengths in the range of 3 to 4 km and horizontal wavelengths along the cross-section in the range of 600 to 700 km. These wavelengths are more than ten times larger than the spatial resolution of the ERA5 data and the gravity waves are therefore well resolvable. In the ERA5 cross-section, we indicated the positions of the cold point for

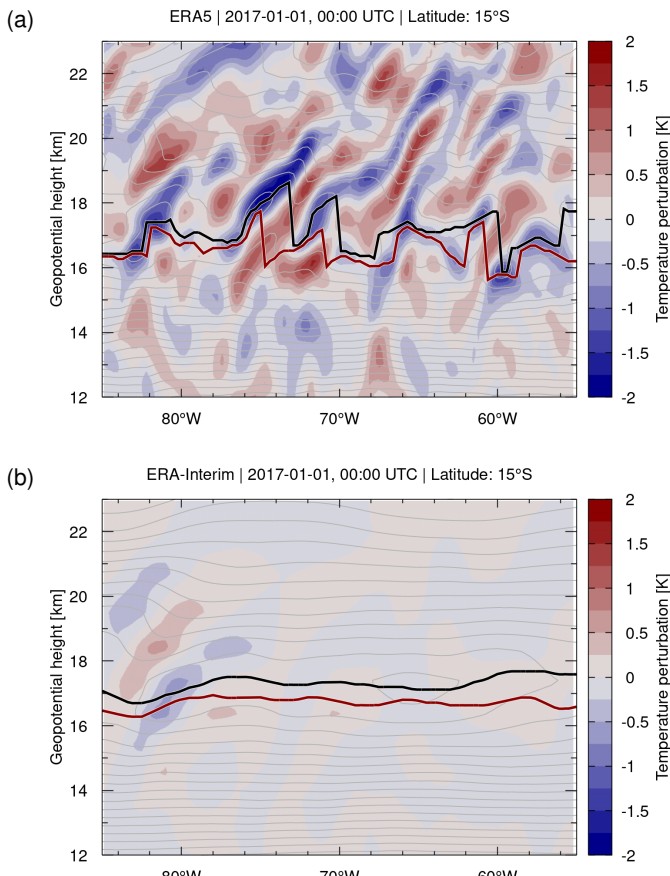

**Figure 10.** Temperature perturbations (contour surface) derived by means of a horizontal Gaussian high-pass filter applied to (a) ERA5 and (b) ERA-Interim reanalysis data on 1 January 2017, 00:00 UTC. Light gray curves are ERA5 and ERA-Interim temperature contours at levels of 2 K, respectively. Also shown are the geopotential heights of the WMO first tropopause (red curve) and the cold point (black curves).

orientation. The locations of the cold point follow the individual cold phase fronts of the gravity waves. The tropopause heights show a similar behavior, although the locations are not directly aligned with the cold wave fronts, as they are defined based on

the temperature gradients according to the WMO lapse rate criteria rather than the temperature minimum. Nevertheless, this example clearly demonstrates that the upper tropospheric and lower stratospheric gravity waves have a pronounced impact on the tropopause structure derived from the ERA5 data. For ERA-Interim, only a much weaker gravity wave event is visible at 80 to 85°W (Fig. 10b), which causes much less variability of the tropopause structure.

Figure 11 shows a comparison of normalized frequency distributions of the temperature fluctuations due to gravity waves

from ERA5 and ERA-Interim. The data refer to 1 January 2017, 00:00 UTC and the 80 hPa level. The calculations were conducted separately for the tropics, mid latitude, and high latitudes, using latitude boundaries at 20 and 65°N/S, respectively. This comparison indicates that the temperature fluctuations at mid latitudes are typically three to five times larger for ERA5



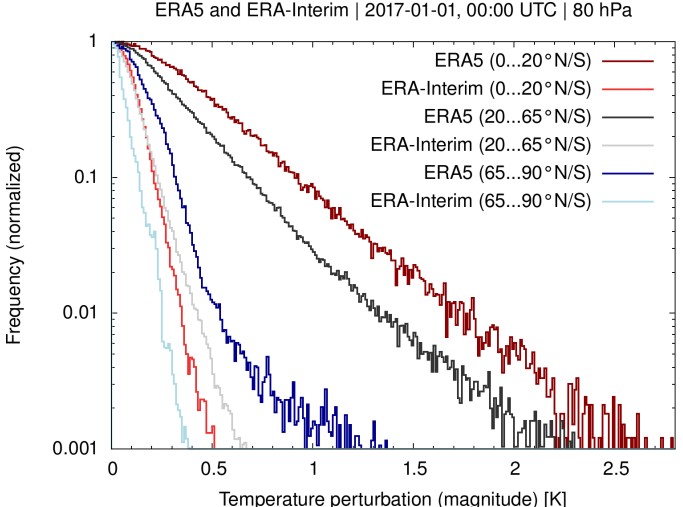

**Figure 11.** Normalized frequency distributions of ERA5 and ERA-Interim temperature fluctuations due to gravity waves on 1 January 2017, 00:00 UTC at the 80 hPa level. The frequency distributions were calculated for the tropics, mid latitudes, and high latitudes, as separated at 20 and 65°N/S, respectively.

than for ERA-Interim. At high latitudes, the differences between ERA5 and ERA-Interim are smaller (about a factor of two to three), but still present. These temperature fluctuations, which are associated with gravity waves in the upper troposphere and

lower stratosphere, are large enough to significantly affect the variability of the tropopause heights and temperatures in ERA5.

### 3.5 Tropopause fluctuations due to convective updrafts

Strong updrafts related to thunderstorms and mesoscale convective systems are another potentially important factor to locally affect tropopause heights and variability. Strong convective updrafts typically lead to lifting of the tropopause, as tropospheric air is transported deeper into the lower stratosphere (Maddox and Mullendore, 2018). Previous comparisons showed that ERA5

better resolves convective updrafts than ERA-Interim, mainly because of its improved spatiotemporal resolution (Hoffmann et al., 2019; Li et al., 2020). For this reason, we investigated whether the updrafts that are explicitly resolved in the ERA5 reanalysis also affect the tropopause.

Convective updrafts can be identified in the reanalysis data in various ways, for instance, based on large vertical velocities, i. e., significant pressure decrease over time, or by inspecting the cloud ice water content (IWC), as strong convective updrafts

and deep convection lead to formation of ice cloud in the upper troposphere. As an example, Fig. 12 shows maps of the ERA5 and ERA-Interim IWC and vertical velocities at 180 hPa (about 12 km of altitude) on 1 January 2017, 00:00 UTC for South America and the neighboring ocean regions. The IWC maps reveal more small-scale variability and local maxima in ERA5 compared to ERA-Interim. Next to gravity waves, a visual inspection of the fine structures in ERA5 reveals convective updrafts in the vertical velocities, which are associated with local increase of the IWC of more than an order of magnitude compared to

ERA-Interim.





**Figure 12.** Ice water content (a,b) and vertical velocities (c,d) of ERA5 (a,c) and ERA-Interim (b,d) on 1 January 2017, 00:00 UTC at the 180 hPa pressure level. The gray box near (45°S, 45°W) highlights a convective event in the ERA5 data (see text for details).

We inspected the ERA5 and ERA-Interim tropopause data for a number of events showing increased IWC and vertical velocities in the upper troposphere due to convective updrafts. Figure 13 highlights a convective event in the ERA5 data near (45°S, 45°W), where the tropopause was uplifted by 1 to 2 km compared to its neighborhood (Fig. 13a). This uplift of the lapse rate tropopause was consistently found also in the dynamical tropopause, as identified by the ±3.5 potential vorticity units (PVU) contour. In contrast, this convective event is not present in the ERA-Interim reanalysis and the tropopause remains flat in the same region (Fig. 13b). As the convective event is present in ERA5 but is completely missing in ERA-Interim, the question arises whether it is actually real. Based on a comparison with the 8.1 micron brightness temperature cloud index, which we calculated from Atmospheric InfraRed Sounder (AIRS) satellite observations (Aumann et al., 2006; Hoffmann and


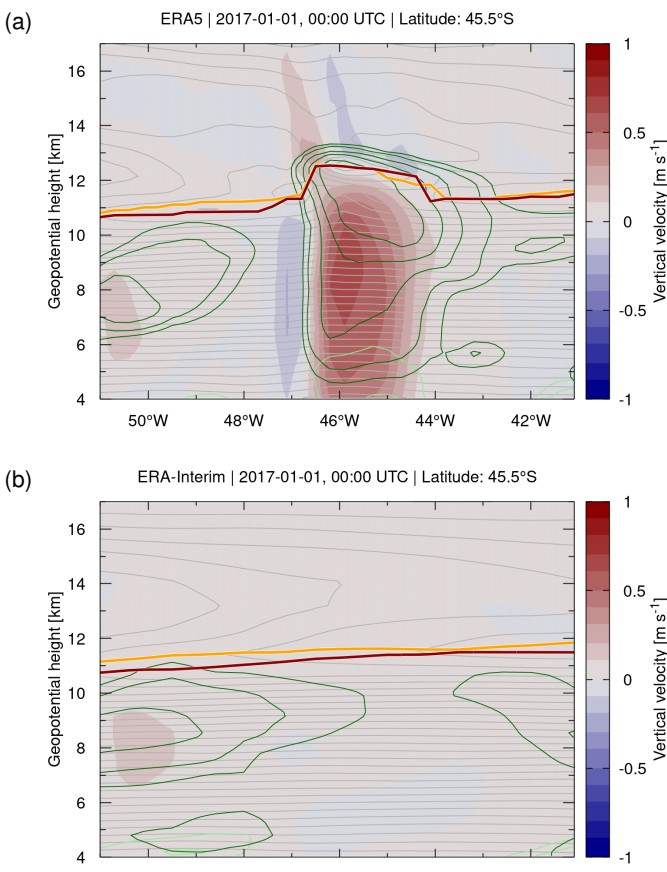

**Figure 13.** Vertical velocities (contour surface) of (a) ERA5 and (b) ERA-Interim reanalysis data on 1 January 2017, 00:00 UTC. Light gray curves show temperature contours at levels of 2 K. Also shown are the geopotential heights of the WMO first tropopause (red curves) and the dynamical tropopause as defined by the ±3.5 PVU contour (orange curves). Dark green and light green curves show contours of the cloud ice and liquid water content at levels of 0.01, 0.1, 1, and 10 g/kg, respectively.

Alexander, 2010), it can be confirmed that the simulated event in ERA5 overlaps with a strong storm system observed in the

study region (see https://datapub.fz-juelich.de/slcs/airs/volcanoes/html/view_2017_001.html, last access: 4 July 2021).

However, although Fig. 13 presents a nice example illustrating how the tropopause can be affected by convection in the ERA5 reanalysis, we found that such events are not very common. From inspecting the five strongest convective updrafts globally on 1 January 2017, 00:00 UTC, the tropopause was affected by convection only for two events. Compared to the extent and effects of gravity waves, we therefore consider the overall impact of convective updrafts on the tropopause as relatively small,

even for the ERA5 reanalysis. Although Hoffmann et al. (2019) found that the vertical velocities of ERA5 in the mid and upper troposphere may exceed those of ERA-Interim by a factor of 2 to 3, the resulting magnitudes of the convective updrafts are still much smaller than those found in real convective updrafts. Convection-permitting simulations would require a horizontal



model resolution on the kilometer-scale whereas the horizontal resolution of ERA5 is limited to 30 km. Convection may play a much larger role in locally affecting the tropopause than deduced here from ERA5 and ERA-Interim.

## 3.6 Sensitivity of tropopause estimates on WMO criteria

The WMO definition (WMO, 1957) provides a set of criteria on how the lapse rate tropopause should be determined from given temperature profiles. However, it remains somewhat unclear on how these criteria were originally established. For this reason, we conducted a statistical analysis of lapse rates found in the height range from the mid troposphere to the lower stratosphere, and we tested the sensitivity of the tropopause height estimates on the lapse rate threshold and the layer depth, in comparison to the standard values of $2\,\mathrm{K\,km^{-1}}$ and 2 km as defined by the WMO.

Figure 14a shows the sensitivity of the ERA5 lapse rate statistics on 1 January 2017, 00:00 UTC over the pressure range from 480 to 60 hPa with respect to the layer depth $\Delta z$. Maximum lapse rates are analyzed here in accordance with the WMO tropopause criterion, which requires that any of the average lapse rates between the tropopause candidate level and the higher levels within $\Delta z$ does not exceed the given lapse rate threshold. The statistics indicate two distinct peaks of maximum lapse rate occurrence frequencies, which are located around 7 to $9\,\mathrm{K\,km^{-1}}$ for tropospheric temperature gradients and $-3$ to $-2\,\mathrm{K\,km^{-1}}$ for lower stratospheric temperature gradients, respectively. Increasing the layer depth $\Delta z$ from 0.1 km to 4 km, the tropospheric and stratospheric peaks become more distinct, suggesting that broad vertical layers are more helpful in separating tropospheric and stratospheric air than narrow layers.

The statistics reveal a minor third peak in lapse rate occurrence frequencies at 2 to $3\,\mathrm{K\,km^{-1}}$. This peak is due to the near isothermal temperature conditions in the polar winter lower stratosphere. The near isothermal conditions and weakened stability in the lower stratosphere are well known to cause difficulties in properly determining the height of the polar winter tropopause (Zängl and Hoinka, 2001). The polar winter peak becomes most pronounced when the statistical analysis is limited to the pressure range from 60 to 120 hPa (about 15 to 20 km) compared to lower levels (Fig. 14b). The winter peak is clearly visible in January and July, but it is mostly suppressed in April and October (Fig. 14c).

The statistical analysis of the lapse rates from the mid troposphere to the lower stratosphere suggests that the thermal tropopause critically depends on the lapse rate threshold and the layer depth applied in the WMO definition. We investigated the sensitivity to these parameters using ERA5 data on 1 January 2017, 00:00 UTC. Figure 15 shows global maps of the geopotential height of the tropopause using lapse rates of 0 and $4\,\mathrm{K\,km^{-1}}$ and layer depths of 0.1 and 4 km, respectively. The results can be compared with Fig. 2a showing the map for the default thresholds defined by the WMO. Figure 16 shows the corresponding zonal means and standard deviations of the tropopause heights on the same day.

Lowering the lapse rate threshold in the range from 4 to $0\,\mathrm{K\,km^{-1}}$ implies an increasingly tight criterion on the separation between the tropospheric and stratospheric temperature gradients and therefore leads to lifting of the mean height of the thermal tropopause between 0.5 km (polar summer high latitudes) and 2 to 3 km at the transition from the tropics to the extratropics (Fig. 16a). At mid latitudes (40 to 60°N/S), the standard deviations of the tropopause heights increase, if the stricter lapse rate threshold of $0\,\mathrm{K\,km^{-1}}$ is applied (Fig. 16b). However, a closer inspection of the maps shows that with the strict threshold, the tropopause algorithm more often tends to fail to detect a tropopause in the winter hemisphere mid and high latitudes (Fig. 15a).



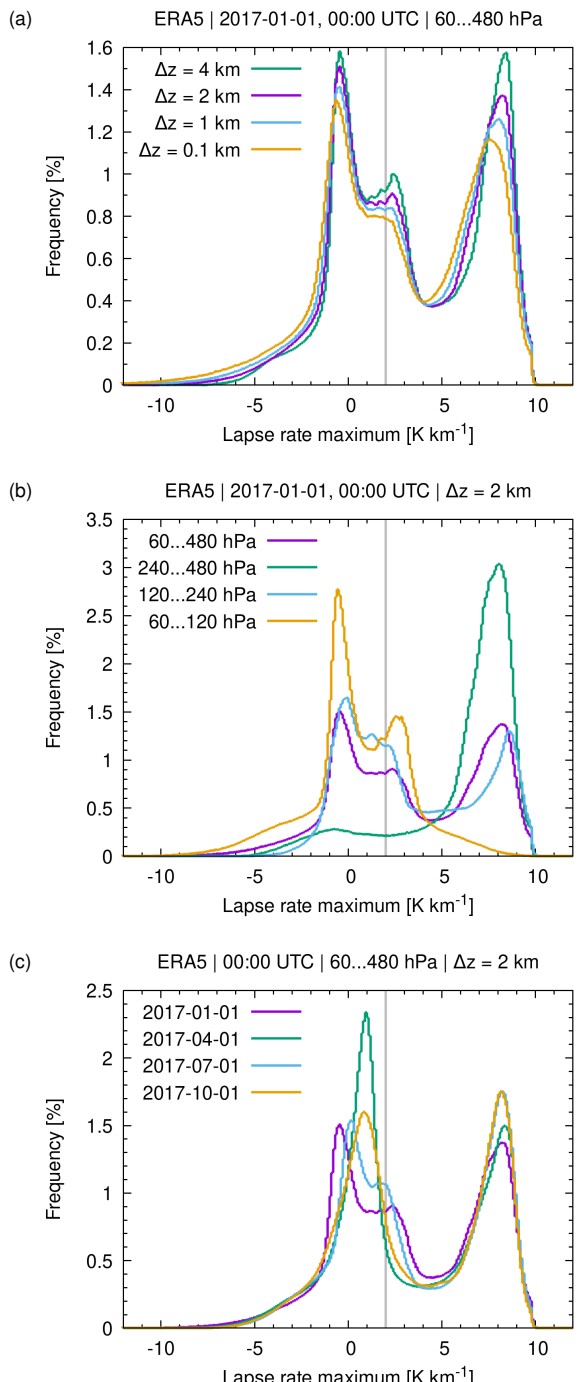

**Figure 14.** Maximum lapse rate statistics from the mid troposphere to the lower stratosphere from ERA5 data (a) for different vertical layer depths $\Delta z$, (b) for different pressure ranges, and (c) on different days. The gray line indicates the $2\,\mathrm{K\,km^{-1}}$ lapse rate threshold of the WMO definition.

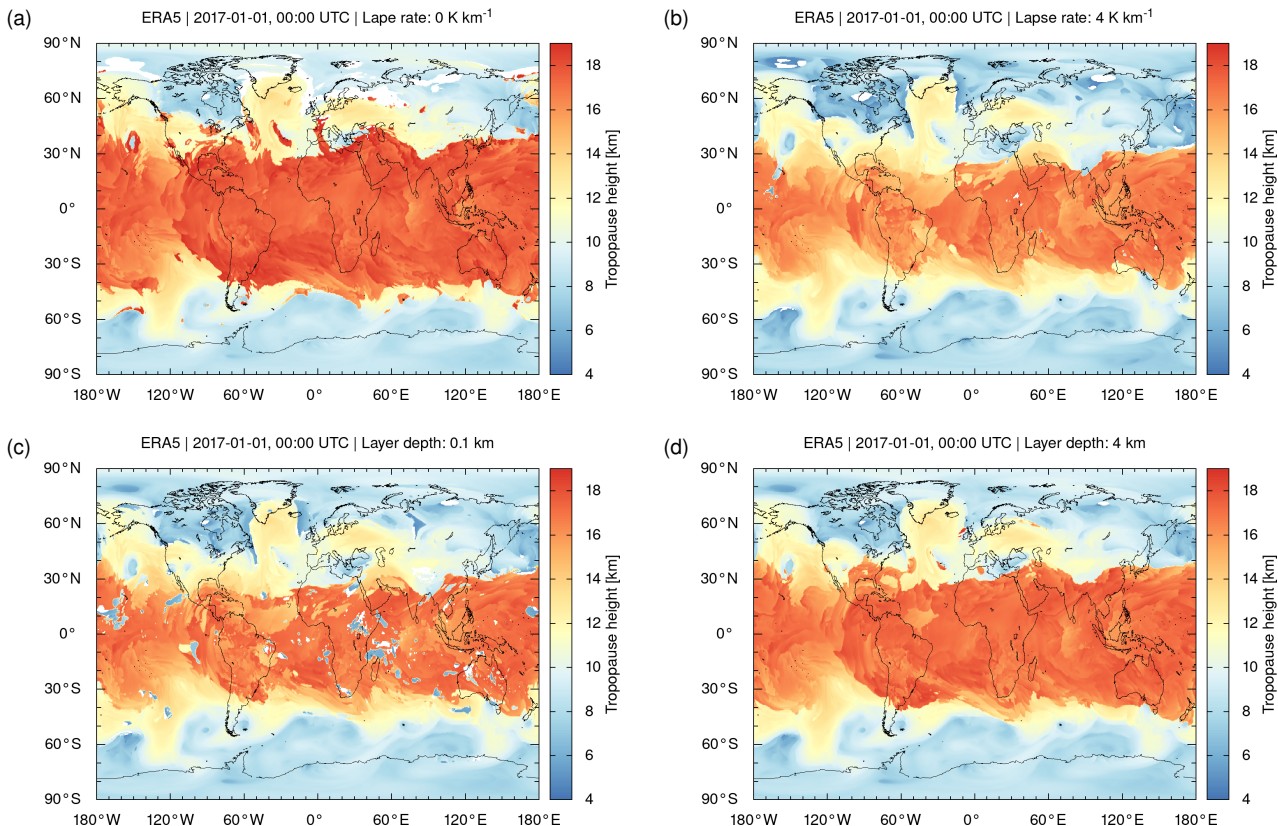

**Figure 15.** Global maps of ERA5 thermal tropopause geopotential heights on 1 January 2017, 00:00 UTC for different lapse rates (a,b) and layer depths (c,d). Compare Fig. 2a for reference.

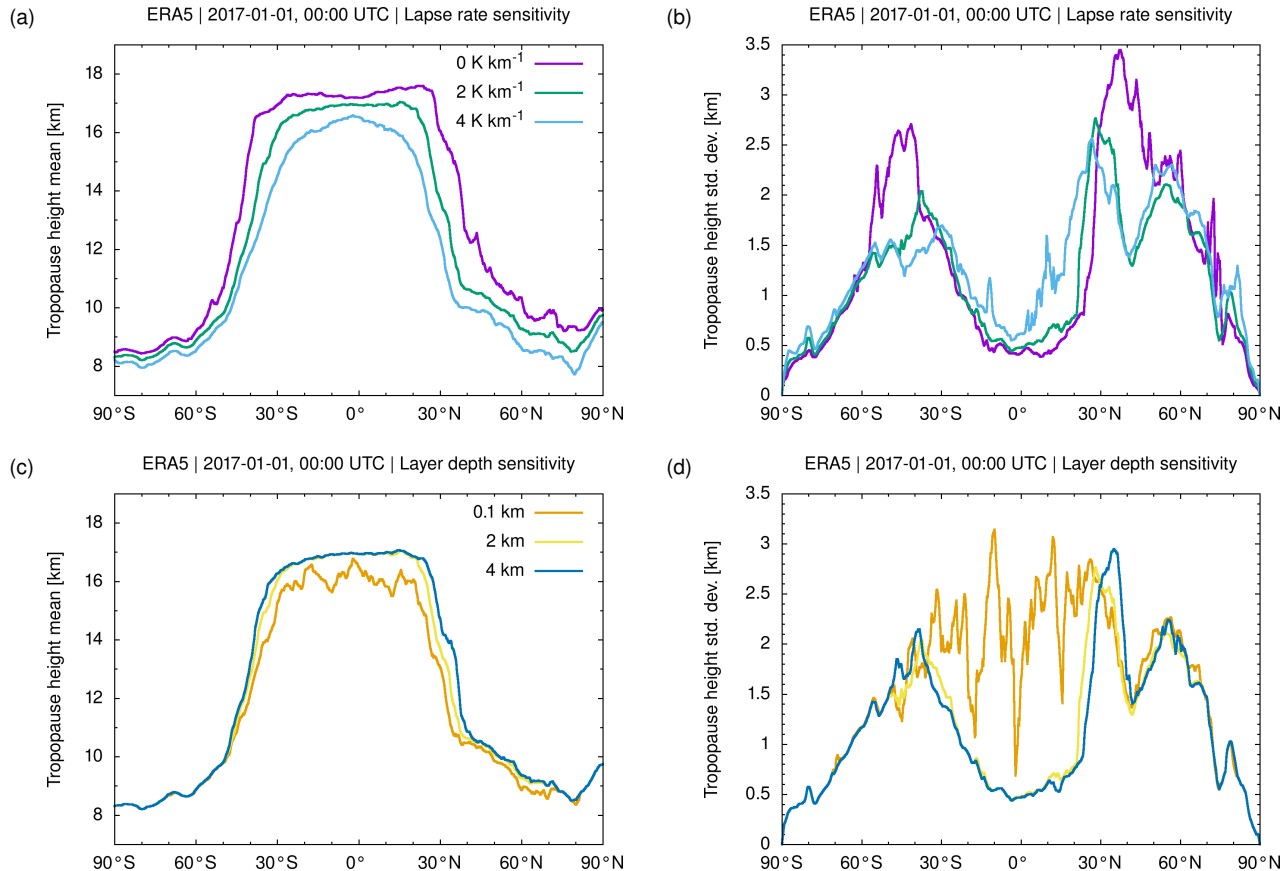

**Figure 16.** Zonal means (a,c) and standard deviations (b,d) of ERA5 thermal tropopause geopotential heights on 1 January 2017, 00:00 UTC. Different curves illustrate the sensitivity of the results on the selected lapse rate threshold (a,b) and the layer depth (c,d).





The sensitivity test on the layer depth mostly reveals difficulties with the tropopause detection in the tropics, if the layer depth is reduced to small values. Here, we tested a layer depth of 0.1 km, which corresponds to the vertical grid spacing applied in the cubic spline interpolation of the temperature profiles. With this threshold, the algorithm identifies tropopause heights as low as $\sim 5\,\text{km}$, which is close to the lower boundary applied for detection. These false detections of the tropopause are found mostly in the tropics and subtropics (Fig. 15c) and relate to aloft thermal inversions within vertically thin layers. The false detections strongly increase the standard deviations of the tropopause heights in the tropics (Fig. 16d). In contrast, increasing the layer depth from 2 to 4 km, the maps and zonal means and standard deviations of the tropopause heights are rather similar compared to the default value of 2 km (Figs. 15d and 16c,d). This implies, our finding that the tropopause height is significantly affected by gravity wave activity in the ERA5 reanalysis is robust, and it is independent of whether a 2 km or a 4 km layer is used to define the tropopause.

### 3.7 Evaluation of tropopause heights with radiosonde data

In this section, tropopause heights from ERA5 and ERA-Interim are compared with HVRRD radiosonde observations. Figure 17 shows a correlation plot of the tropopause heights. For the first tropopause (TPH1) the correlation is better than for the second tropopause (TPH2) for both reanalyses (0.989/0.986 for ERA5/ERA-Interim for TPH1 and 0.608/0.628 for THP2, respectively). It is surprising that the correlation coefficients show no substantial improvement when moving from ERA-Interim to ERA5, or are even better for ERA-Interim than ERA5, like for TPH2. However, for TPH2, the correlations for ERA5 are slightly lower than ERA-Interim due to two interacting effects: (a) the different number of TPH2 detections (712 for ERA5 / 484 for ERA-Interim), which is a noteworthy result on its own; and (b) the differences in TPH2 are generally larger than for TPH1, whereby outliers in the range of a few kilometers are more common. Figure 17 suggests that the reanalyses have a tendency to overestimate TPH2, especially due to the outliers. It should be noted that the total number of TPH2 events in the radiosonde data is substantially higher (2106 events, see also Tab. 1) compared to both reanalyses. These numbers highlight the general effect, that the enhanced vertical resolution of ERA5 yields a higher probability to detect a secondary tropopause, but the resolution is still not sufficient to detect many of the TPH2 events in the HVRRD data.

A couple of statistics showing the over- or underestimation of the reanalysis tropopauses with respect to the *truth*, the radiosonde and GPS results, are summarized in Table 1. The first tropopause has a clear tendency to be overestimated compared to the *truth*, with similar percentages for ERA5 (63.3/36.7 % over-/underestimated tropopause events) and ERA-Interim (62.6/37.4 %). For the second tropopause, the tendency is switched, although the imbalance is smaller. A second measure tries to quantify the tendency to "larger" discrepancies by counting only events with tropopause height differences ($\text{dTPH}_1$) larger than $\pm 200\,\text{m}$ and second tropopause height differences ($\text{dTPH}_2$) larger than $\pm 400\,\text{m}$. Here, a larger threshold for TPH2 was selected to account for the larger variability of TPH2 compared to TPH1. The computation of TPH2 is more sensitive to the vertical resolution and sampling issues of the measurements or model input than the first tropopause. Table 1 shows that a significant overestimation of the true TPH1 is less frequent for ERA5 (30.7 %) than for ERA-Interim (37.0 %). It also shows a similar tendency to underestimate TPH1 although the absolute values are significantly smaller (13.5 and 18.2 %, respectively). In summary, ERA5 reduced the issue of over- or underestimating TPH1 and TPH2 with respect to HVRRD.





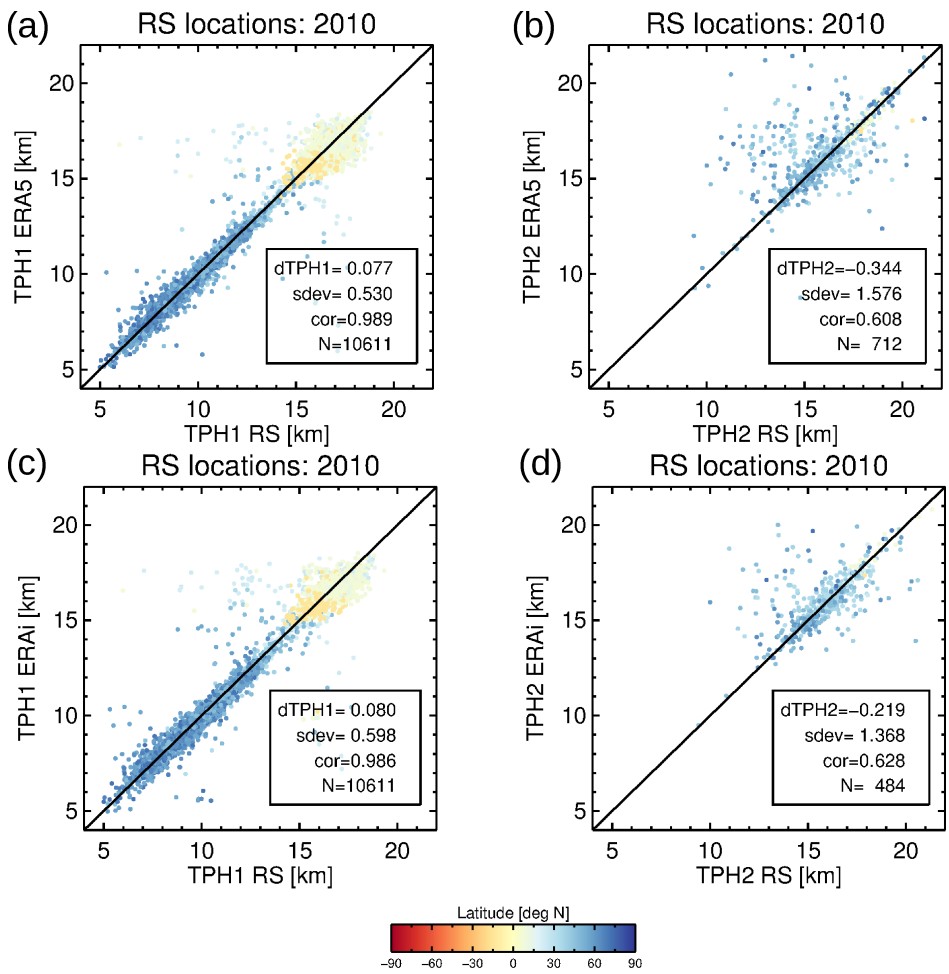

**Figure 17.** Correlation diagram of HVRRD first (a,c) and second (b,d) tropopause heights versus ERA5 (top) and ERA-Interim (bottom) tropopause heights for the 2010 radiosonde (RS) locations, respectively.

Finally, we applied a percentile analysis to the tropopause height differences. The HVRRD is not a global data set but covers a certain latitude-longitude range with good temporal and horizontal sampling. Zonal mean statistics of the differences between HVRRD and ERA5/ERA-Interim tropopause heights for the $P_{10}$, $P_{25}$, $P_{50}$ (median), $P_{75}$, and $P_{90}$ percentiles were calculated and are presented in Fig. 18. Percentiles are better suited than mean and standard deviation to highlight asymmetries and tendencies in the distributions. For $\mathrm{dTPH_1}$, both reanalyses show a small positive bias in the median (as already expected from Table 1), ERA5 has $\simeq 50\,\%$ smaller median difference along the 20° latitude bins than ERA-Interim. For both reanalyses, the percentiles show a symmetric behavior with respect to the median values, with a stronger mean asymmetry (positive) for ERA-Interim than for ERA5. Obviously, the uncertainty in the determination of the first tropopause becomes larger at tropical latitudes, and this effect is even more pronounced for ERA-Interim than for ERA5.





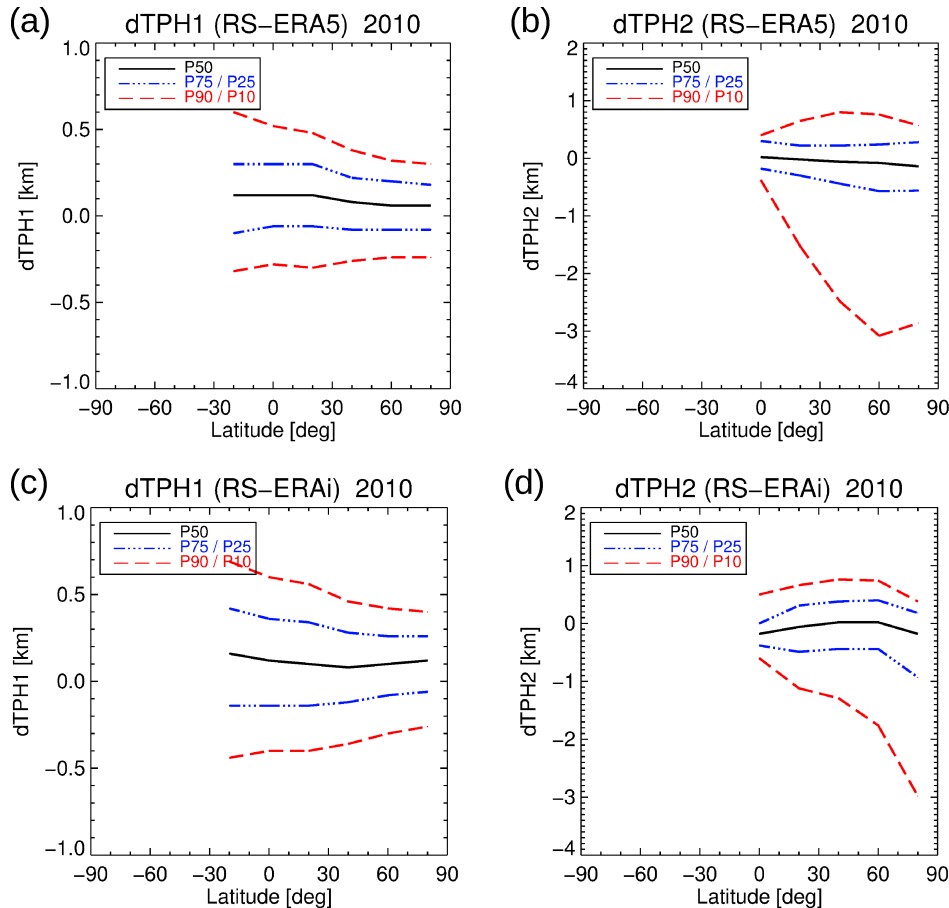

**Figure 18.** Zonal percentiles $P_{10}$, $P_{25}$, $P_{50}$ (median), $P_{75}$, and $P_{90}$ of the tropopause height differences between HVRRD and ERA5 for the first tropopause ($\mathrm{dTPH_1}$) in (a) and the second tropopause ($\mathrm{dTPH_2}$) in (b), and for ERA-Interim in (c) and (d), respectively. Note the change and asymmetry in the y-range between first and second tropopause differences. Statistics apply for the year 2010.

For the second tropopause, the differences are generally larger, but still 50 % of the measurements ($P_{75} - P_{25}$) show a difference of less than $\simeq$400 m. For higher latitudes the negative differences are increasing to up to 3 km, especially for ERA5. There are indications that the temperature structure at high/mid latitudes in winter and autumn (isothermal structure in the lower stratosphere) leads to overestimated and potentially false detection of secondary tropopause events. Under such conditions, ERA5 is producing still more second tropopause events than ERA-Interim, but also a larger negative bias in $\mathrm{dTPH_2}$ for the
high latitudes.

### 3.8    Evaluation of tropopause heights with GPS data

As described in Sect. 2.3, the GPS radio occultation measurements compiled in the COSMIC project constitute the second data set for evaluation. Here, a nearly global coverage and a factor of $\simeq$50 times more profiles than for the HVRRD data can





**Table 1.** Summary of tropopause height differences between ERA5 and ERA-Interim with radiosonde (RS) and GPS profiles.

| | $dTPH_1 >0$ m [%] | $dTPH_1 <0$ m [%] | $N_{ERA}$ / $N_{RS}$ | $dTPH_2 >0$ m [%] | $dTPH_2 <0$ m [%] | $N_{ERA}$ / $N_{RS}$ / [%] |
|---|---|---|---|---|---|---|
| RS − ERA5 | 63.3 | 36.7 | 10611 / 10611 | 44.2 | 55.8 | 712 / 2106 / 33.8 |
| RS − ERA-Interim | 62.6 | 37.4 | 10611 / 10611 | 48.1 | 51.9 | 484 / 2106 / 23.0 |
| | $dTPH_1 >200$ m [%] | $dTPH_1 < -200$ m [%] | | $dTPH_2 >400$m [%] | $dTPH_2 < -400$ m [%] | |
| RS − ERA5 | 30.7 | 13.5 | | 16.3 | 26.4 | |
| RS − ERA-Interim | 37.0 | 18.2 | | 21.9 | 27.2 | |
| | $dTPH_1 >0$ m [%] | $dTPH_1 <0$ m [%] | $N_{ERA}$ / $N_{GPS}$ | $dTPH_2 >0$ m [%] | $dTPH_2 <0$ m [%] | $N_{ERA}$ / $N_{GPS}$ / [%] |
| GPS − ERA5 | 56.8 | 43.2 | 561,665 / 561,666 | 52.1 | 48.8 | 60,139 / 109,363 / 55.0 |
| GPS − ERA-Interim | 59.3 | 40.1 | 562,568 / 563,040 | 49.3 | 50.7 | 49,028 / 109,677 / 44.7 |
| | $dTPH_1 >200$ m [%] | $dTPH_1 < -200$ m [%] | | $dTPH_2 >400$m [%] | $dTPH_2 < -400$ m [%] | |
| GPS − ERA5 | 18.2 | 13.2 | | 12.5 | 16.1 | |
| GPS − ERA-Interim | 29.5 | 17.9 | | 16.0 | 19.9 | |

be taken into account in a 1-year comparison. Due to the large number of profiles, the presentation of the probability density

function (PDF) in Fig. 19 is better suited than the scatter plot used for the radiosonde comparison. The comparison between ERA5 and ERA-Interim results shows a narrower spread for ERA5 than ERA-Interim for both, TPH1 and TPH2. Obviously, TPH1s between 13 and 15 km are less frequent than at the altitudes below and above. TPH1s below 13 and above 15 km show also somewhat larger spread with respect to the reference data. The large total number of GPS observations (>500,000) gives a high confidence for this conclusion. Tropopause altitudes between 13 and 15 km are most frequently observed in the

subtropical jet regions, where the transition between the high tropical (> 14 km) and low mid-latitude (< 12 km) tropopause occurs and double tropopause events are frequently observed (c. f., Figs. 5a and 8a). Note that the PDFs for TPH1 suggest a broad scatter for the tropopause height differences, but it should be taken into account that most of the dark blue areas in the PDFs is due to single events. Large discrepancies of several kilometers for TPH1 are possible for both reanalyses, but the majority of the distribution is close (±200 m) to the one-to-one line (see Table 1).

At first glance, the PDFs for TPH2 look similar for both reanalyses. However, the better vertical resolution of ERA5 results in a finer sampling (in the order of the bin sizes), whereby for ERA-Interim the sampling looks more distinct. A step-like behavior appears for ERA-Interim above 15 km of altitude, with steps in the range of the vertical resolution of ERA-Interim (about 1 to 1.2 km). It should be stressed again that the ERA5 data set comprises significantly more second tropopause events than the ERA-Interim data set (60,139 compared to 49,028), but the fraction of second tropopause events resolved by the reanalyses

is larger for GPS than for the radiosondes (ERA5: 55.0 % / 33.8 % and ERA-Interim: 44.7 % / 23.0 %). Very likely, the much better vertical resolution for the radiosondes allows the detection of second tropopause events, which are not detectable with neither the reanalyses nor the GPS profiles. Consequently, these statistics result in a larger consistency between the reanalyses and GPS compared to the radiosondes.





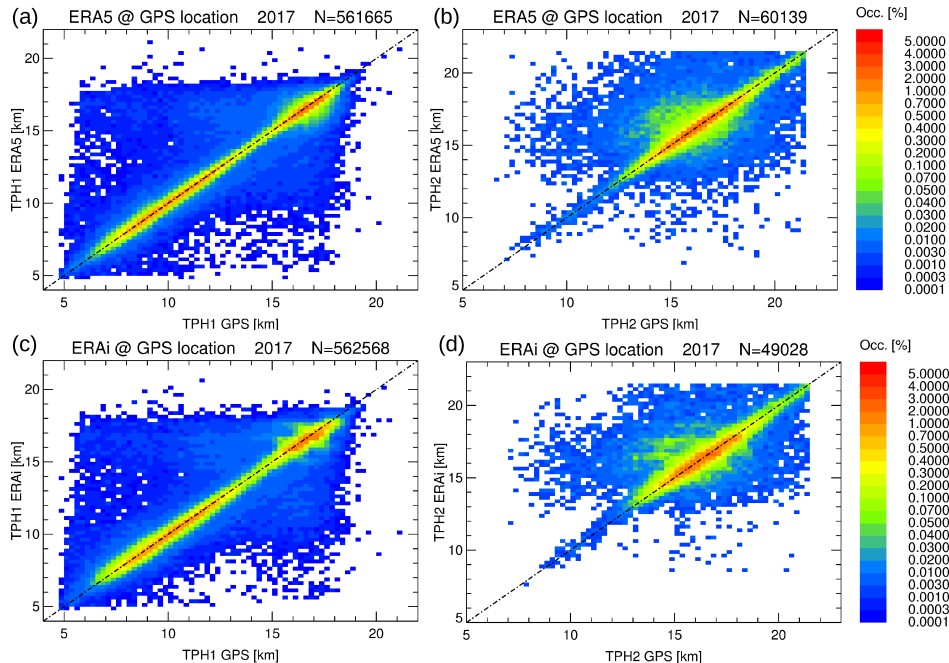

**Figure 19.** Probability density functions of first (a,c) and second (b,d) tropopause heights from GPS versus ERA5 (a,b) and ERA-Interim (c,d) for all COSMIC profiles in 2017. The bin size is 200 m × 200 m. Note the log-like intervals of the color bar. For details see text.

The percentile analysis of the tropopause height differences is summarized – like already for the radiosonde data – in Table 1. The coarser vertical resolution of the GPS measurements compared to the radiosondes results in a smaller asymmetry with respect to the under- and overestimation for $TPH_1$ ($dTPH_1 < 0$ m and $dTPH_1 > 0$ m, respectively) for ERA5 and ERA-Interim. It is slightly better for ERA5 than ERA-Interim. There is nearly no asymmetry for $dTPH_2$ with values close to 50 %, which is equivalent to a median close to zero for both reanalyses.

The latitude dependence of the tropopause differences with respect to GPS is presented in Fig. 20. For the first tropopause, the results for the different percentiles are are symmetric with respect to the equator. Enhanced values for $P_{90}$ can be observed in the subtropics and jet regions 20a,c. There is a slightly larger spread for the southern polar region than for the northern polar region for the first tropopause. The $P_{50}$ highlights with values close to zero the non-biased character of the distribution, also the other percentiles are rather symmetric to the zero line. However, the spread of $P_{75} - P_{25}$ and $P_{90} - P_{10}$ is smaller for ERA5 than for ERA-Interim.

For the second tropopause (Fig. 20b,d), the ERA5 results show a smaller spread and slightly smaller latitudinal values for $P_{75}$ and $P_{25}$ than ERA-Interim. A pronounced change of the percentiles in the subtropics like for the first tropopause is not detectable. Similar as for the northern high latitudes in the radiosonde comparison, positive outliers (overestimated TPH2 with respect to GPS) can be identified for northern and southern high latitudes, to some extent already at mid latitudes. Whether these areas with large $dTPH_2$ spread are related to the special thermal structure of the polar lower stratosphere in winter, was





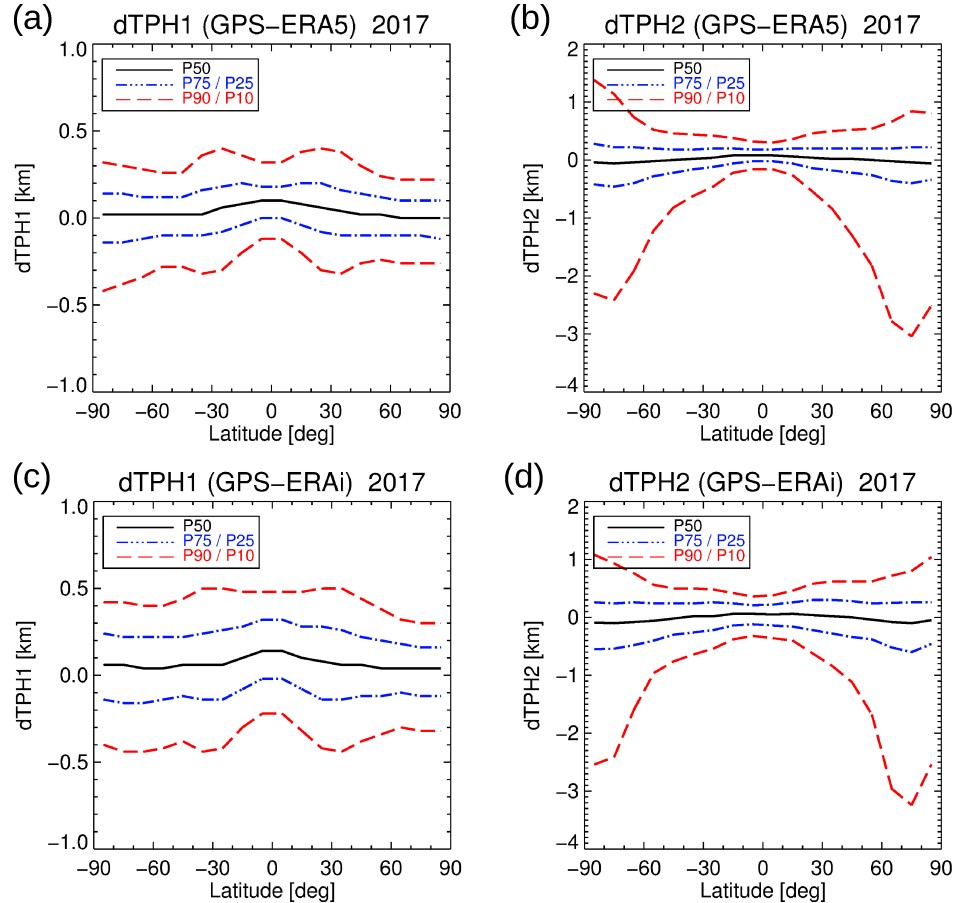

**Figure 20.** Zonally percentiles $P_{10}$, $P_{25}$, $P_{50}$ (median), $P_{75}$, and $P_{90}$ of tropopause height differences in 2017 between GPS and ERA5 (top) and ERA-Interim (bottom), respectively, with delta tropopause height ($\mathrm{dTPH_1}$) for the first (a,c) and $\mathrm{dTPH_2}$ for the second lapse rate tropopause following the WMO criterion (b,d). The same axis ranges as in Fig. 18 are used for $\mathrm{dTPH_1}$ and $\mathrm{dTPH_2}$.

investigated by a simple approach. By eliminating all profiles for latitudes $> 50°$N/S in Nov-Mar and Apr-Sep, respectively, the new statistics show a large reduction for $P_{10}$ to values smaller than 1 km (see Fig. C1 in Appendix C). This analysis also shows that April is already a month when strong differences in the second tropopause determination appear that need to be excluded to reduce the differences of the second tropopause heights between the reanalyses and GPS data.

Finally, we investigated a potential temporal change in the differences of GPS and the reanalyses by an identical analysis
for the year 2010. PDF statistics and percentile analysis show only minor differences to year 2017. Consequently, temporal changes in the quality of the tropopause parameters of the reanalyses over a timescale of several years are unlikely.





## 4 Conclusions

In this study, we conducted an assessment of the characteristics of the tropopause based on ten-year records (2009 – 2018) of the ERA5 and ERA-Interim meteorological reanalyses. In contrast to the large-scale features, our study found significant dif-
ferences between the ERA5 and ERA-Interim tropopause characteristics on smaller scales, in particular in the tropics. Monthly standard deviations of tropopause heights of ERA5 are up to 60 % larger than those of ERA-Interim and monthly standard deviations of tropopause temperatures are up to 30 % larger. This larger variability of the ERA5 tropopause heights and temperatures is considered realistic and attributed to improved spatiotemporal resolution and better representation of geophysical processes in the ECMWF forecast model as well as to improvements in the data assimilation scheme and utilization of addi-
tional observations in ERA5. In particular, the improved spatiotemporal resolution of ERA5 allows for better representation of mesoscale features such as gravity waves and convective updrafts, which significantly affect the temperature profiles in the upper troposphere and lower stratosphere and therefore also affect the tropopause characteristics.

Overall, although the variability and gravity wave-induced fluctuations in the ERA5 temperature profiles are more pronounced than in the ERA-Interim data set, the retrieved tropopause heights, first and second, show a better and more compact
correspondence with GPS and high-resolution radiosonde observations. This suggests that better vertical resolution plus more realistic temperature fluctuations are important parameters for the accuracy of the tropopause determination based on reanalysis data. In the present study, ERA5 shows better performance than ERA-Interim on estimating the first and second tropopause height with respect to both reference data sets. The comparison indicates an uncertainty of the first tropopause for ERA5 (ERA-Interim) of about $\pm 150$ m to $\pm 200$ m ($\pm 250$ m) based on HVRRD data and $\pm 120$ m to $\pm 150$ m ($\pm 170$ m to $\pm 200$ m) based on
the coarser resolution GPS data at different latitudes.

Numerous studies demonstrated that mesoscale numerical simulations are well capable of explicitly representing gravity waves from sources such as orography, convection, or jet and storm sources (Plougonven et al., 2013; Orr et al., 2015; Alexander et al., 2016; Stephan et al., 2019a,b; Heale et al., 2020). These studies showed that the horizontal and vertical resolution of the numerical models as well as different filters applied for numerical stability are key factors in determining the fraction of the
gravity wave spectrum that can be explicitly resolved in the simulations. Other studies evaluated how realistically explicitly resolved gravity waves are represented in operational analyses and reanalyses in comparison to observations (Schroeder et al., 2009; Jewtoukoff et al., 2015; Hoffmann et al., 2017a,b; Lambert and Santee, 2018). These studies showed that the spatial and temporal patterns of the occurrence of gravity wave events are well reproduced in reanalyses. However, depending on the spatiotemporal resolution of the forecast model, reanalyses will typically underestimate real gravity wave amplitudes.
Other gravity wave characteristics such as the horizontal and vertical wavelengths or the phase of the waves may also not be reproduced exactly.

Nevertheless, newer reanalyses from state-of-the-art forecast models and data assimilation systems with improved spatiotemporal resolution are generally expected to have better representation of explicitly resolved gravity waves. This is in line with results of the present study, showing that ERA5 has more small-scale variability in tropopause heights due to gravity
wave-induced temperature fluctuations than ERA-Interim. Despite much larger variability, the comparison of the reanalysis



tropopause heights with HVRRD and GPS observations indicates that ERA5 better represents real tropopause height charac-
teristics than ERA-Interim. Therefore, ERA5 will be an important asset for future research activities related to the tropopause.

*Data availability.* The ERA5 and ERA-Interim reanalysis data sets (Dee et al., 2011; Hersbach and Dee, 2016) were retrieved from ECMWF's
Meteorological Archival and Retrieval System (MARS). See https://www.ecmwf.int/en/forecasts/datasets/browse-reanalysis-datasets (last
access: 18 November 2021) for further details. The ERA5 and ERA-Interim reanalysis tropopause data sets presented in this paper are
made available via a data repository hosted at https://datapub.fz-juelich.de/slcs/tropopause/ (last access: 18 November 2021) or via https:
//doi.org/10.26165/JUELICH-DATA/UBNGI2 (last access: 18 November 2021). The GPS satellite data are distributed by the COSMIC Data
Analysis and Archive Center (CDAAC) via there website at https://cdaac-www.cosmic.ucar.edu/cdaac (last access: 18 November 2021).
The US HVRRD are available from the SPARC website at https://www.sparc-climate.org/data-centre/data-access/us-radiosonde/ (last ac-
cess: 18 November 2021). The corresponding GPS and HVRRD tropopause data used in this study can be obtained from Reinhold Spang
(r.spang@fz-juelich.de).

## Appendix A:  Comparison of model level and pressure level based tropopause estimates

As part of the data processing, we derived the tropopause heights from ECMWF reanalysis data sets that were interpolated
from IFS model levels to pressure levels (Sect. 2.1). The vertical interpolation of the geopotential heights and temperature
profiles potentially introduces interpolation errors. To estimate the interpolation errors, we compared the tropopause heights
from the pressure level data with tropopause heights that were derived independently and directly from model level data.

Figure A1 shows global maps of ERA5 and ERA-Interim tropopause heights on 1 January 2017, 00:00 UTC from the
reanalysis model level data and the corresponding differences with respect to the pressure level data (cf. Fig. 2). For the
majority of the data, this comparison reveals very good agreement between the pressure and model level results. However,
two exceptions should be noted. First, at the transitions from the tropics to the extratropics and in other regions with strong
tropopause gradients, locally quite large differences (more than $\pm 1$ km) may occur between the pressure level and model level
data. Second, ERA-Interim tropical tropopause heights from the pressure level data exhibit a systematic low bias compared to
the model level data. For the ERA5 data, this low bias is less pronounced.

In order to quantify the differences between the pressure level and model level data, we calculated the median, the interquar-
tile range, and the difference of the $P_{10}$ and $P_{90}$ percentiles of the tropopause height differences. Considering that the statistical
distributions of the differences shown in Fig. A1 are largely non-Gaussian, we decided to present these more robust statistics
here, as they are less sensitive to outliers compared to mean and standard deviation. Overall, this analysis indicates very good
agreement between the pressure and model level tropopause data. The zonal median of the height differences is in the range of
$-40$ to $0$ m for ERA5 and $-100$ to $0$ m for ERA-Interim. The interquartile range is 50 to 70 m for ERA5 and 50 to 250 m for
ERA-Interim. The $P_{90} - P_{10}$ difference is about 80 to 130 m for ERA5 and 100 to 500 m for ERA-Interim. These differences
are much smaller than the vertical resolution of the reanalysis data sets. Therefore, we conclude that the vertical interpolation
from model level to pressure level data does not introduce any large uncertainties in our analysis.





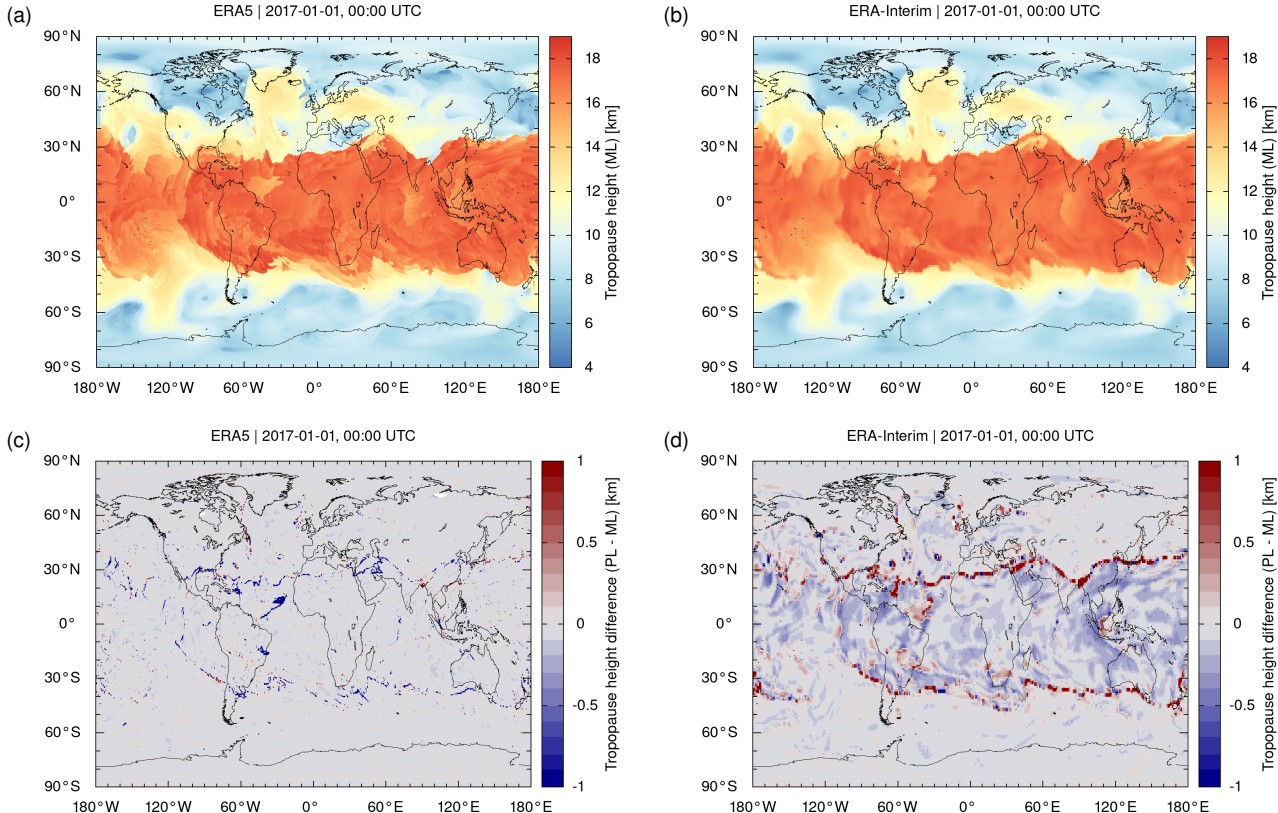

**Figure A1.** Global maps of tropopause geopotential heights from (a) ERA5 and (b) ERA-Interim as derived from reanalysis model level (ML) data. The corresponding pressure level (PL) data are shown in Fig. 2. Differences between pressure and model level data (PL − ML) are shown in (c) and (d), respectively. Note that the color bar ranges for the differences are restricted to ±1 km, to make smaller differences visible. The data refer to 1 January 2017, 00:00 UTC.

## Appendix B: Comparison of linear and cubic spline interpolation for tropopause determination

In our data processing, we applied a cubic spline interpolation to transfer the pressure, temperature, and geopotential height
profiles from the coarser resolution reanalysis levels to a finer vertical grid (Sect. 2.2). We tested, how the results compare to a simpler solution, using linear interpolation instead of cubic spline interpolation. Figure B1 compares ERA-Interim tropopause geopotential heights on 1 January 2017, 00:00 UTC from the linear interpolation and the cubic spline interpolation.

This comparison shows that with the linear interpolation method, the tropopause height estimates remain confined to the coarser resolution vertical spacing of the reanalysis data. This follows, as linear interpolation of the temperature profile implies
a constant lapse rate between the reanalysis levels. But with a constant lapse rate between the coarse levels, the WMO criterion does not help to more accurately identify the tropopause on the refined grid. The application of the cubic spline method is therefore considered essential to improve the vertical resolution of the tropopause heights with our method.





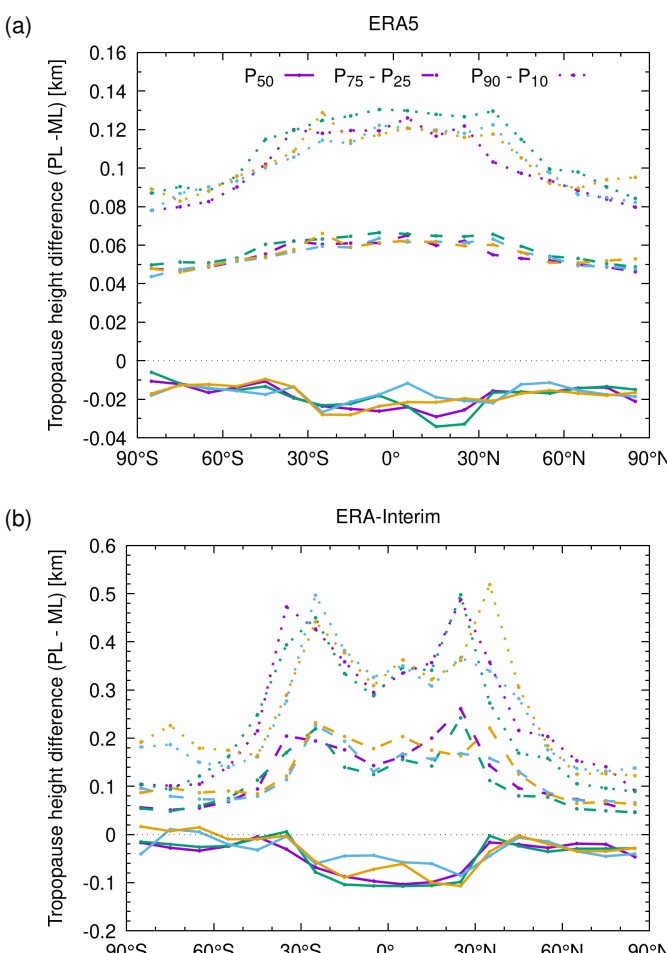

**Figure A2.** Zonal differences of tropopause geopotential heights derived from model level (ML) and pressure level (PL) data for (a) ERA5 and (b) ERA-Interim. Different curves show the median ($P_{50}$), the interquartile range ($P_{75} - P_{25}$), and the difference of the $P_{90}$ and $P_{10}$ percentiles of the differences. Colors refer to different days (purple: 1 January 2017, green: 1 April 2017, blue: 1 July 2017, orange: 1 October 2017). Data are shown for 00:00 UTC each day.

Note that Reichler et al. (2003) proposed an alternative method to estimate tropopause heights from coarser resolution gridded data. In their method, Reichler et al. (2003) apply linear interpolation to the lapse rates between the mid-levels of the gridded data. Their method allows for more accurate determination of the tropopause heights, but it should also be noted that linear interpolation of the lapse rates implies second-order interpolation of the temperature profiles. Therefore, higher-order interpolation of the temperature profile is also a key element of the method of Reichler et al. (2003) to improve the vertical resolution of the tropopause height estimates, similar to the cubic spline method applied here.





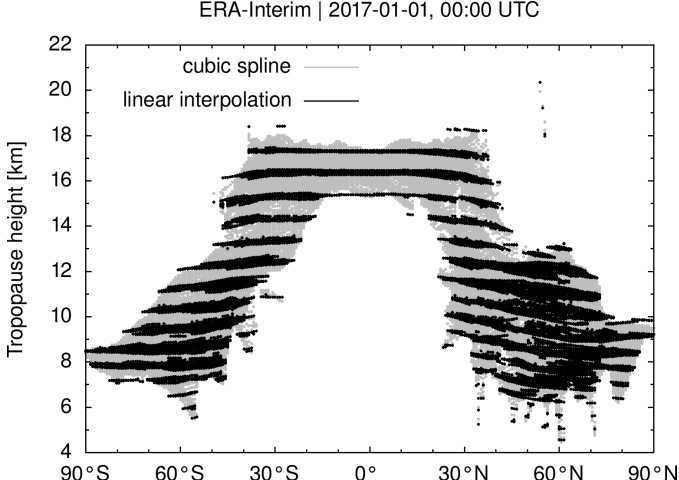

**Figure B1.** Tropopause geopotential heights from ERA-Interim data for 1 January 2017, 00:00. The tropopause heights were estimated by applying linear interpolation (black dots) and cubic spline interpolation (gray dots) to refine the vertical temperature profiles of the reanalysis data.

## Appendix C:  A no-winter tropopause height difference statistic

The unexpected large discrepancies of the second tropopause of the reanalyses with respect to GPS and radiosondes (Fig. 20 and 18) at polar latitudes are potentially caused by the isothermal temperature structure in the lower stratosphere in polar winter conditions. For a simple test of this hypothesis, we tried to exclude different months from the statistics to suppress the influence of the polar winter profiles. The final result is presented in Fig. C1, where all profiles north and south of $\pm 50°$ of latitude in November to March and April to September are excluded, respectively. For the Southern Hemisphere, April looks

not like winter but is quite a crucial month for the improvement. Only the exclusion of April results in a substantial reduction of the percentile $P_{10}$ south of $50°$S form $> 2\,\mathrm{km}$ to $< 1\,\mathrm{km}$.

## Appendix D:  Reanalysis tropopause data repository

In this study, we estimated tropopause data from more than $6.3 \times 10^{10}$ vertical profiles of pressure, temperature, and geopotential height of the $2009 - 2018$ ERA5 record. Considering the effort to process the large amount of ERA5 data, we decided to make

the reanalysis tropopause data derived here available for future studies via a dedicated data repository (Hoffmann and Spang, 2021). The data repository is accessible at https://datapub.fz-juelich.de/slcs/tropopause/ (last access: 18 November 2021) or via https://doi.org/10.26165/JUELICH-DATA/UBNGI2 (last access: 18 November 2021). In addition to ERA-Interim and ERA5 data, the repository provides access to tropopause estimates from the Modern-Era Retrospective Analysis for Research and Applications, Version 2 (MERRA-2, Gelaro et al., 2017) and the NCEP/NCAR Reanalysis-1 (Kalnay et al., 1996). The

repository currently provides data for the years 2009 to 2018 for all records, but data processing is ongoing and the repository





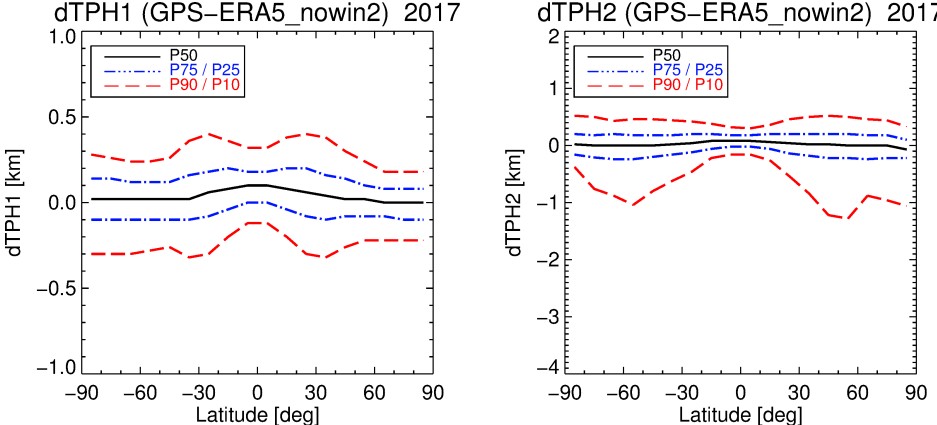

**Figure C1.** Zonally percentiles of $P_{10}$, $P_{25}$, $P_{50}$ (median), $P_{75}$, and $P_{90}$ of the difference in tropopause heights between GPS and ERA5 for 2017, first (left) and second lapse rate tropopause (right) and non polar-winter conditions (for details see text).

will eventually be extended to cover the complete time periods of the different reanalysis data sets. All data sets were processed with the same algorithms as described in this paper for comparability and consistency. The data are provided in terms of daily netCDF files. In addition to WMO first and second tropopause data, the data files also provide pressure, temperature, geopotential heights, and humidity of the cold point and the dynamical tropopause (based on thresholds of 3.5 potential vorticity

units in the extratropics and 380 K of potential temperature in the tropics). The website of the data repository provides access to browse images of daily ERA5 tropopause data. Access to the repository is open and the data are provided under a Creative Commons Attribution 4.0 International License. We hope that this open resource will become a valuable asset in future research on the manifold dynamical and chemical processes at the tropopause.

*Author contributions.* LH and RS jointly developed the concept of this study. LH prepared the ECMWF reanalysis tropopause data sets.
RS prepared the GPS and radiosonde tropopause data sets. Both authors conducted the data analysis, contributed to the interpretation of the results, and wrote the manuscript.

*Competing interests.* The authors declare that no competing interests are present.

*Acknowledgements.* This work was supported by the German Research Foundation (DFG) through the project AeroTrac under the grant ID: DFG HO5102/1-1. ERA5 data were generated using Copernicus Climate Change Service Information. Neither the European Commission nor
ECMWF are responsible for any use that may be made of the Copernicus information or data in this publication. We acknowledge the Jülich



Supercomputing Centre for providing computing time and storage resources on the supercomputers JUWELS. We thank our colleagues at the Institute of Energy and Climate Research and at the Jülich Supercomputing Centre for providing helpful feedback and suggestions.



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
