# Peer review of "An assessment of tropopause characteristics of the ERA5 and ERA-Interim meteorological reanalyses"

_Atmospheric Chemistry and Physics, 2021_

## Referee Comment (RC2)

**Review of "An assessment of tropopause characteristics of the ERA5 and ERA-Interim meteorological reanalyses" by Lars Hoffmann and Reinhold Spang.**

This paper describes the differences in tropopause (lapse-rate tropopause following the WMO definition) characteristics between the European Centre for Medium-Range Weather Forecasts (ECMWF) reanalysis data ERA-5 and its predecessor ERA-Interim based on data for the 2009-2018 period. The tropopause in ERA-5 is on average 150 m (in the tropics) to 300 m (around 30°) lower and colder by 1.5K compared to ERA-Interim. It is also shown that the tropopause field in ERA5 exhibits a larger variability which is attributed to the higher horizontal/vertical resolution of the ECMWF model used to produce the reanalysis data (30 km for ERA-5 versus 80 km for ERA-Interim, ~500 m in the Upper Troposphere-Lower Stratosphere versus ~1 km for ERA-Interim). As a result mesoscale processes such as gravity waves which play a large role in tropopause variability are better represented in the more recent ERA5 data. Further comparison of the reanalysis data with GPS satellite (COSMIC, MetOp) and radiosonde observations indicates that ERA 5 tropopause height has a lower difference (around 150m) with the obs than ERA-Interim and is thus more suitable for tropopause-related studies.

The paper is interesting and well written. The paper is suitable for publication in ACP. I would suggest providing more information to answer the following minor comment:

It is explained that the tropopause field in ERA5 has a larger variability because of a better representation of mesoscale processes due to the higher horizontal (~30km)/vertical (500m in the UTLS) resolutions used in the Integrated Forecasting System (IFS) model used to produce the ERA5 reanalyses. However, 30 km is still relatively coarse to resolve deep convection, which needs to be parameterized in the IFS model. In the tropics, where convection is the main source of waves, is it possible to disentangle the impact of data assimilation versus better representation of mesoscale processes on the representation of the tropopause? Especially, since you are focusing on a recent period (2009-2018) where more GPS satellite observations became available.

The higher vertical resolution in the UTLS certainly helps to better represent the wave vertical propagation and effects on the temperature field but maybe you could comment a bit more on the fact that the source of the wave is still partially resolved and that data assimilation may play a larger role? Have you considered comparisons of ERA5 with other modern reanalyses (e.g. MERRA2).

---

## Author Comment (AC1)

**Reply to review comments**

We thank the reviewers and the editor for the time and effort spent on the manuscript and for providing helpful comments. We considered all comments and hope that the revised draft properly addresses the open issues. Please find our point-by-point replies below (colored in blue). A revised manuscript with tracked changes has been uploaded.

**Reviewer #1**

This is an excellent tropopause analysis comparing the two most recent versions of ECMWF reanalysis. The exploration of the data and convincing demonstration of the factors leading to differences and quality of tropopause heights was varied, creative, and convincingly done. Comparisons between the reanalyses and several high-resolution observational datasets were key to the quality estimation. It was a pleasure reviewing this study. I only have a handful of minor points of constructive criticism for the authors to consider as they revise the paper.

We would like to thank the reviewer a lot for this encouraging statement.

I would recommend using the common acronym UTLS throughout in place of "upper troposphere and lower stratosphere"

Revised as suggested.

Line 159: nearest neighbor is really a sampling approach rather than "interpolation"

Rephrased as "nearest neighbor sampling".

Figure 2: would be nice to have the vertical section paths of Figs. 3 & 4 overlaid here

We added lines indicating the positions of the meridional and zonal cross-sections in Figs. 2 and 7, respectively.

First part of Section 3.2 (mainly lines 234-250): I would recommend calling out the common LRT errors near the S pole here rather than later in the paper (currently at lines 395-397). There is currently too much focus here assuming these tropopause ranges and characteristics appropriately characterize the UTLS in this region.

We agree and moved the description of the issues in determining the LRT in polar winter conditions from lines 395-397 forward to Sect. 3.2.

Line 287: I am uncomfortable with the assertion that the double tropopause identifications are an "artifact" here. If they satisfy the WMO criteria, how can they be an artifact? Perhaps what is intended here is a distinction between the various formation mechanisms for double tropopauses. In the literature cited in the paper (and in other studies not cited), there are known dynamic pathways for double tropopause formation (including

lateral transport from Rossby wave breaking near the subtropical jets and column stretching/shrinking in the UTLS) and contributions from radiation.

We agree with the reviewer, that these events fulfil the WMO criteria and should not be titled as artifacts. The temperature structure is like assimilated by the model and is resulting in a double tropopause location even if the distance between the first and second tropopause looks unrealistically large. In addition, a downward moving cold polar vortex can be considered as a further mechanism creating a second tropopause in addition to the dynamical pathways you listed above. Consequently, we changed the text accordingly.

Line 355: recommending revising to "...lead to detrainment and formation of ice cloud..."

Rephrased as suggested.

Line 473: "is due to" should be "are due to"

Fixed.

**Reviewer #2**

This paper describes the differences in tropopause (lapse-rate tropopause following the WMO definition) characteristics between the European Centre for Medium-Range Weather Forecasts (ECMWF) reanalysis data ERA-5 and its predecessor ERA-Interim based on data for the 2009-2018 period. The tropopause in ERA-5 is on average 150 m (in the tropics) to 300 m (around 30°) lower and colder by 1.5K compared to ERA-Interim. It is also shown that the tropopause field in ERA5 exhibits a larger variability which is attributed to the higher horizontal/vertical resolution of the ECMWF model used to produce the reanalysis data (30 km for ERA-5 versus 80 km for ERA-Interim, 500 m in the Upper Troposphere-Lower Stratosphere versus 1 km for ERA-Interim). As a result mesoscale processes such as gravity waves which play a large role in tropopause variability are better represented in the more recent ERA5 data. Further comparison of the reanalysis data with GPS satellite (COSMIC, MetOp) and radiosonde observations indicates that ERA5 tropopause height has a lower difference (around 150 m) with the obs than ERA-Interim and is thus more suitable for tropopause-related studies.

The paper is interesting and well written. The paper is suitable for publication in ACP.

We would like to thank the reviewer for this supportive statement.

I would suggest providing more information to answer the following minor comment: It is explained that the tropopause field in ERA5 has a larger variability because of a better representation of mesoscale processes due to the higher horizontal ( 30 km)/vertical (500 m in the UTLS) resolutions used in the Integrated Forecasting System (IFS) model used to produce the ERA5 reanalyses. However, 30 km is still relatively coarse to resolve deep convection, which needs to be parameterized in the IFS model. In the tropics, where

convection is the main source of waves, is it possible to disentangle the impact of data assimilation versus better representation of mesoscale processes on the representation of the tropopause? Especially, since you are focusing on a recent period (2009-2018) where more GPS satellite observations became available.

We agree that this is an important question. In Sect. 3.5 of the paper, we concluded that the resolution of ERA5 is still too coarse to explicitly resolve deep convection and associated uplifts of the tropopause. As this information was missing in the discussion of the paper, we added a new paragraph: "Deep convection is expected to affect tropopause heights as it causes uplifting of air from the troposphere towards the stratosphere (e.g. Maddox and Mullendore, 2018). In earlier work, we found that ERA5 better represents tropospheric updrafts than ERA-Interim (Hoffmann et al., 2019) but the present study shows that uplifts of the tropopause due to deep convection are not very common in both, ERA5 and ERA-Interim. In future work, it would be interesting to inspect the tropopause characteristics in convection-permitting simulations with horizontal model resolution on the kilometer-scale. Such data sets became recently available, for instance, within the DYnamics of the Atmospheric general circulation Modeled On Non-hydrostatic Domains (DYAMOND) project (Stevens et al., 2019; Stephan et al., 2019a,b). Convection may play a much larger role in locally affecting the tropopause than deduced here from the ECMWF reanalyses. In addition, it would be interesting to assess forecasts in contrast to reanalyses, to learn more about the effects of data assimilation of high-resolution observations on the mesoscale variability of the tropopause."

The higher vertical resolution in the UTLS certainly helps to better represent the wave vertical propagation and effects on the temperature field but maybe you could comment a bit more on the fact that the source of the wave is still partially resolved and that data assimilation may play a larger role? Have you considered comparisons of ERA5 with other modern reanalyses (e.g. MERRA2).

We agree it would be very interesting to study the effects of data assimilation on the mesoscale variability of the tropopause in the reanalysis data sets. With our present data and methods, this is unfortunately not easily possible, and we would have to leave this question for a future study. In the discussion section, we added this suggestion, please see reply to previous comment.

**Additional changes**

We added a second affiliation for both authors.

**References**

Hoffmann, L., Günther, G., Li, D., Stein, O., Wu, X., Griessbach, S., Heng, Y., Konopka, P., Müller, R., Vogel, B., and Wright, J. S.: From ERA-Interim to ERA5: the considerable impact of ECMWF's next-generation reanalysis on Lagrangian transport simulations, Atmos. Chem. Phys., 19, 3097–3124, doi: 10.5194/acp-19-3097-2019, 2019.

Maddox, E. M. and Mullendore, G. L.: Determination of Best Tropopause Definition for Convective Transport Studies, J. Atmos. Sci., 75, 3433 – 3446, doi: 10.1175/JAS-D-18-0032.1, 2018.

Stephan, C. C., Strube, C., Klocke, D., Ern, M., Hoffmann, L., Preusse, P., and Schmidt, H.: Gravity Waves in Global High-Resolution Simulations With Explicit and Parameterized Convection, J. Geophys. Res., 124, 4446–4459, doi: 10.1029/2018JD030073, 2019a.

Stephan, C. C., Strube, C., Klocke, D., Ern, M., Hoffmann, L., Preusse, P., and Schmidt, H.: Intercomparison of Gravity Waves in Global Convection-Permitting Models, J. Atmos. Sci., 76, 2739 – 2759, doi: 10.1175/JAS-D-19-0040.1, 2019b.

Stevens, B., Satoh, M., Auger, L., Biercamp, J., Bretherton, C. S., Chen, X., Düben, P., Judt, F., Khairoutdinov, M., Klocke, D., et al.: DYAMOND: the DYnamics of the Atmospheric general circulation Modeled On Non-hydrostatic Domains, Prog. Earth Planet. Sci., 6, 1–17, 2019.